# Conjunctive spatial and self-motion codes are topographically organized in the GABAergic cells of the lateral septum

**Suzanne van der Veldt**, **Guillaume Etter**, **Coralie-Anne Mosser**, **Frédéric Manseau**, **Sylvain Williams** *

McGill University & Douglas Mental Health University Institute, Montreal, Canada

* sylvain.williams@mcgill.ca

## Abstract

The hippocampal spatial code's relevance for downstream neuronal populations—particularly its major subcortical output the lateral septum (LS)—is still poorly understood. Here, using calcium imaging combined with unbiased analytical methods, we functionally characterized and compared the spatial tuning of LS GABAergic cells to those of dorsal CA3 and CA1 cells. We identified a significant number of LS cells that are modulated by place, speed, acceleration, and direction, as well as conjunctions of these properties, directly comparable to hippocampal CA1 and CA3 spatially modulated cells. Interestingly, Bayesian decoding of position based on LS spatial cells reflected the animal's location as accurately as decoding using the activity of hippocampal pyramidal cells. A portion of LS cells showed stable spatial codes over the course of multiple days, potentially reflecting long-term episodic memory. The distributions of cells exhibiting these properties formed gradients along the anterior–posterior and dorsal–ventral axes of the LS, directly reflecting the topographical organization of hippocampal inputs to the LS. Finally, we show using transsynaptic tracing that LS neurons receiving CA3 and CA1 excitatory input send projections to the hypothalamus and medial septum, regions that are not targeted directly by principal cells of the dorsal hippocampus. Together, our findings demonstrate that the LS accurately and robustly represents spatial, directional as well as self-motion information and is uniquely positioned to relay this information from the hippocampus to its downstream regions, thus occupying a key position within a distributed spatial memory network.

## Introduction

The lateral septum (LS) is an anatomically complex structure uniquely positioned within a broader distributed spatial memory network. The LS consists almost exclusively of highly interconnected inhibitory GABAergic cells [1,2] that are densely innervated by the principal cells of the hippocampus [3,4], in contrast to the medial septum (MS), which receives inputs solely from the inhibitory neurons of the hippocampus [5]. Previous research has implicated the LS in a wide variety of behaviors, including spatial and working memory [6–10], regulation of feeding [11–14], anxiety [15–18], locomotion [19,20], and social behaviors [21–24].

**Data Availability Statement:** All relevant quantitative data are within the paper and its Supporting Information files. Code for cell extraction is available at https://github.com/

etterguillaume/MiniscopeAnalysis. Code for behavior extraction is available at https://github.com/etterguillaume/PIMPN. Code for extraction of tuning curves and decoding can be downloaded at the following address: https://github.com/etterguillaume/CaImDecoding. All other code is available at https://github.com/suzannevdveldt/spatialcodingLS.

**Funding:** This work was supported by funding from the Canadian Institutes for Health Research (CIHR) Foundation Program FDN-148478, the Natural Sciences and Engineering Research Council of Canada (NSERC) Discovery Grant RGPIN-2020-06717, and a Tier 1 Canada Research Chair to SW. SVDV is supported by a Vanier Canada Graduate Scholarship and the Richard H. Tomlinson Doctoral Fellowship. C-AM is supported by a FRQS postdoctoral fellowship. The funders had no role in study design, data collection and analysis, decision to publish, or preparation of the manuscript.

**Competing interests:** The authors have declared that no competing interests exist.

**Abbreviations:** GRIN, gradient refractive index; LH, lateral hypothalamus; LS, lateral septum; MI, mutual information; MS, medial septum; VTA, ventral tegmental area.

To date, functional characterizations of the LS have often focused on the spatial coding properties of these neurons, primarily due to the extensive excitatory input from positionally tuned hippocampal pyramidal neurons, commonly referred to as "place cells" [4,24–26]. Yet, this body of work has not yet yielded a consensus. Several studies have reported place-like cells in the LS, but the number, information content, and stability vary widely from study to study [20,27–31], while others described a lack of canonical place cells altogether [4]. As previous studies often depended on variable criteria of what defines a place cell, estimates range from 5.3% [4] to 56.0% [20] of LS cells classified as spatially modulated. Strikingly, even estimates by the same authors on data acquired from the same subjects engaging the same task ranged between 26.5% [32] to 56.0% [20]. When reported, LS place cells are typically described as of lesser quality than classic hippocampal place cells, with lower within-session stability and larger place fields [30,31]. Adding to this complexity, the LS is a large structure that is cytoarchitecturally subdivided into dorsal, intermediate, and ventral subregions that spread across its anterior–posterior axis [5,26,33]. This suggests that, in addition to subjective criteria, the striking differences in LS spatial coding characteristics between studies may be in part a product of variations in recording locations within the region. In addition to positional information, LS neurons may also encode direction and self-motion information, including acceleration and velocity [20,30,31].

Beyond their spatial tuning properties, hippocampal place cells are also noted for the rapid reorganization of the place fields across time (on the order of days) and navigational contexts, a phenomenon termed "remapping" [34–38]. If the LS spatial code is directly inherited from its hippocampal inputs, one would expect comparable dynamics over a similar timescale. While the LS, similar to the hippocampus, has been implicated in behaviors that may require a stable representation of space over time, including context-reward [39,40], contextual fear conditioning [16,41–44], spatial learning and memory [32,45,46] remapping dynamics in the LS remain largely understudied. Thus, whether and how the LS spatial code changes across time is crucial to clarifying its relationship to the upstream hippocampal spatial code, with important implications for its involvement in this broader spatial memory network and how it supports behaviors across longer timescales.

To resolve these outstanding questions, we functionally characterize hippocampal connectivity of LS and asked how information-rich LS GABAergic neurons are in relation to their main pyramidal inputs in dorsal CA3 and CA1. To this end, we used head-mounted miniaturized microscopes to record approximately 2,000 LS neurons across the anterior–posterior and dorsal–ventral axes of the LS, and compared their firing characteristics to pyramidal cells recorded in CA1 and CA3, as animals navigated linear track and open field environments. Results demonstrate that the LS accurately and robustly represents spatial, speed, and directional information in an anatomically organized fashion. While the information of LS neurons is generally comparable to CA3 and CA1, our tracing studies suggest that LS information is sent to downstream brain regions such as the MS and hypothalamus that are not directly connected by the principal cells of the dorsal hippocampus.

## Results

### LS GABAergic neurons exhibit stable spatial activity during alternation on a linear track

Previous electrophysiological studies of the LS disagree on both the proportion of place encoding cells in this region and their spatial content. Cells have been previously described to be of lower quality than the classically described hippocampal place cells [20,30,32] or to be virtually absent [4].

First, to assess the distribution of excitatory pyramidal inputs from each hippocampal sub-region to the GABAergic cells of the LS, we injected a monosynaptic retrograde rabies tracer in dorsal LS (Figs 1A and S1, $n$ = 5 mice), utilizing the monosynaptically restricted, Cre-dependent retrograde tracing features of this technique. We observed strong labeling of pyramidal neurons of the hippocampus in areas CA1 and CA3 (Figs 1B and S1). To investigate how the spatial coding properties of the LS compare to these main dorsal hippocampal inputs, we expressed the calcium ($Ca^{2+}$) indicator GCaMP6f in GABAergic cells of the LS or in the pyramidal cells of the dorsal CA1 (dCA1) and dorsal CA3 (dCA3) of mice (Figs 1C, S2 and S3). We recorded $Ca^{2+}$ activity from 1,082 LS GABAergic neurons ($n$ = 15 mice), 1,251 dCA1 pyramidal cells ($n$ = 5 mice), and 464 cells from dCA3 ($n$ = 6 mice) from animals alternating on a linear track. We subsequently filtered and binarized the recorded calcium traces (S4A Fig) [47] and excluded periods of immobility ($<5$ cm.s$^{-1}$). Using an information-theoretic framework [48–50], we computed the mutual information (MI) between calcium activity and spatial location for each cell and expressed the results in bits (S4B Fig). As we observed a significant correlation between probability of a cell to be active (P(A)) and MI (S4B Fig), we computed the MI value for 30 bootstrap samples (50% data sampling) and compared those to 30 circularly shuffled surrogates, which allowed us to compute significance of the MI value independently of P(A) (S4C–S4E Fig) [47]. We found that LS GABAergic activity was significantly spatially modulated ($p < 0.01$, Figs 1D and S4H). Indeed, both MI and within-session stability were higher for spatially modulated LS cells compared to nonmodulated cells (Mann–Whitney test, $p < 0.0001$, S4E and S4F Fig). Using this approach, a total of 43.90% of LS GABAergic cells qualified as spatially modulated (475/1,082 cells), with LS cells similarly distributed along the linear track as place cells recorded from dorsal CA1 and dorsal CA3 (Fig 1E). In contrast to previous reports [4,20,32], we observed a significant difference in the proportion of spatially modulated cells across recording regions (LS: 43.57 ± 4.46%; dCA1: 29.60 ± 2.99%; dCA3: 27.01 ± 7.07%; Kruskal–Wallis, H(3) = 6.212, $p$ = 0.0448; Dunn's multiple comparisons test: LS-dCA1: $p$ = 0.1757; LS-dCA3: $p$ = 0.1101; Fig 1F). We found that within-session stability of spatially modulated cells (expressed as a correlation where 0 means no stability while 1 means perfectly stable) was significantly different between LS (0.44 ± .0165) and dCA3 (0.73 ± 0.26; Kruskal–Wallis, H(3) = 80.48, $p < 0.0001$; Dunn's multiple comparisons test, $p < 0.0001$), and between dCA1 and dCA3 ($p < 0.0001$; Fig 1G). Even though LS GABAergic cells are not fast spiking (S5 and S6 Figs) [51], P(A) in LS (0.02005 ± 4.692 × 10$^4$) is higher than both dCA1 (0.0094 ± 2.544 × 10$^4$, Kruskal–Wallis, H(3) = 502.0, $p < 0.0001$; Dunn's multiple comparisons test, $p > 0.0001$) and dCA3 (0.009225 ± 3.759 × 10$^4$, $p < 0.0001$, S6A and S6B Fig). Therefore, we assessed the MI in bits carried in each binarized event and observed that the LS (5.624 × 10$^5$ ± 1.151 × 10$^6$ bits/binarized event) carries significantly less information per binarized event than CA1 (1.430 × 10$^4$ ± 1.613 × 10$^6$ bits/binarized event; Kruskal–Wallis, H(3) = 1,288, $p < 0.0001$; Dunn's multiple comparisons test, $p < 0.0001$); and CA3 (1.130 × 10$^4$ ± 2.507 × 10$^6$; bits/binarized event; $p < 0.0001$; Fig 1H). To confirm that the observed differences in proportion of spatially modulated cells and within-session stability were not due to differences in P(A), we used an activity cutoff of P(A) > 0.001 and confirmed our previous results (S6D Fig).

In contrast to their omnidirectional firing properties in open field environments, hippocampal neurons rapidly develop strong directional selectivity on linear tracks [52–54]. To test whether the LS place code developed comparable directional selectivity, we ranked spatially modulated cells as a function of the difference between MI for left and right traversals (S7A–S7D Fig), and found LS cells tuned to the absolute location regardless of direction (7.67%, 83/1,082 cells), while other cells were selective to the left (15.52%, 168/1,082 cells) or right trajectories (22.08%, 239/1,082 cells).

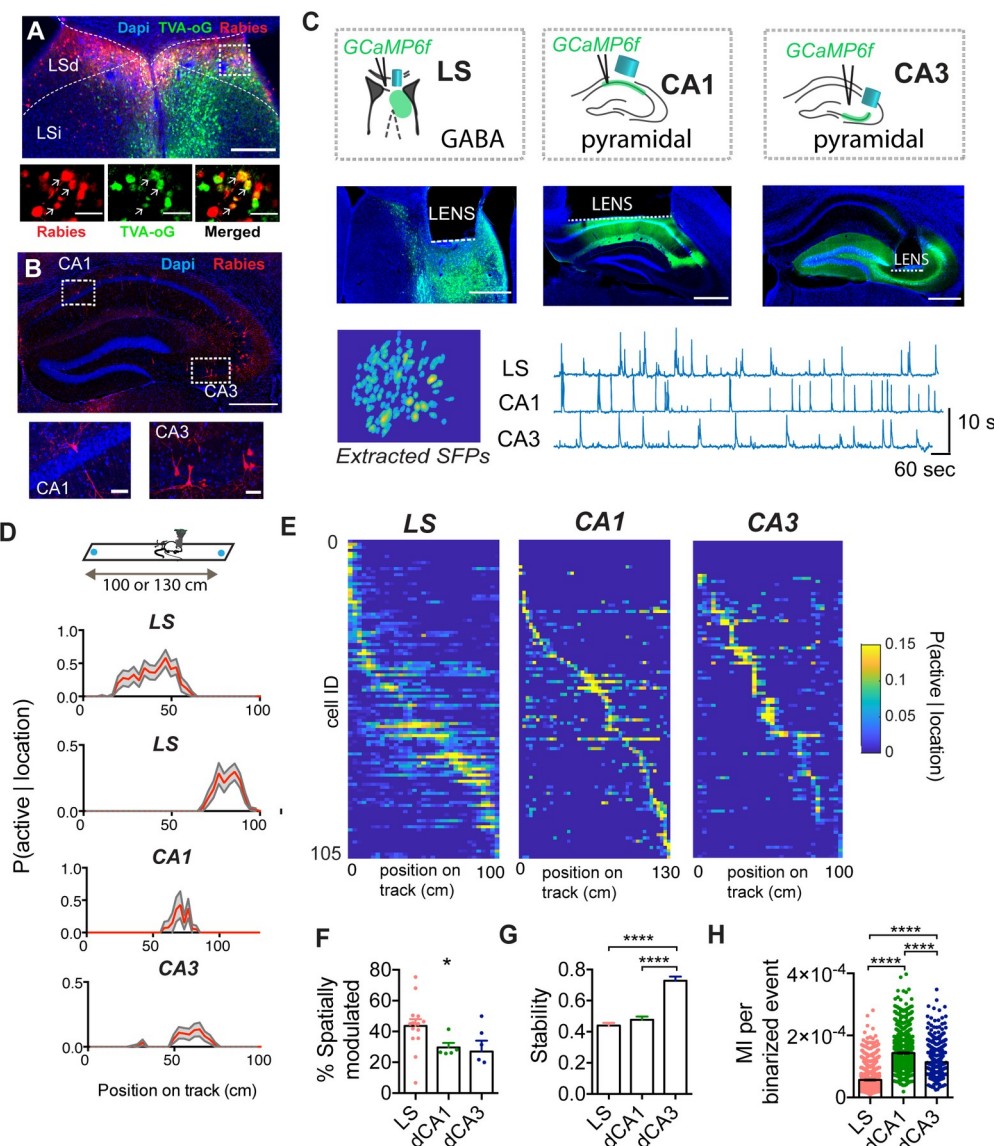

**Fig 1. LS shares spatial coding characteristics with its main hippocampal inputs CA1 and CA3 during goal directed navigation in a 1D environment.** (A) Retrograde rabies tracing injection site, blue: DAPI staining, green: TVA.oG coupled to an eGFP, red: Rabies coupled to mCherry, orange: starter cells expressing both TVA.oG.eGFP and Rabies.mCherry. (B) Retrograde labeling in the dorsal hippocampus, coronal section. Bottom left: example of mCherry-positive CA1 pyramidal cell, bottom right: mCherry-positive CA3 pyramidal cells, blue: DAPI staining, red: Rabies coupled to mCherry. (For additional images, see S1 Fig). (C) Top: diagram of one-photon calcium recording setup in LS, CA1, and CA3 in freely behaving mice, with GCaMP6f expression restricted to GABAergic cells in LS and restricted to pyramidal cells in dorsal CA1 and CA3, middle: histological verification of implantation site (see also S2 and S3 Figs), and bottom: extracted calcium transients for LS, CA1, and CA3. (D) Top: linear track paradigm, with sucrose rewards on either end. Bottom: probability of an example cell to be active given the location on the linear track (red) with 95% upper and lower percentile (gray). Examples shown are from 2 different LS mice and a representative spatially modulated cell from one CA1 and CA3 animal each. (E) Activity of cells sorted along location in the maze for each region (blue, low; yellow, high). (F) Percent of significantly spatially modulated cells for each animal for each recording region. LS: $N$ = 15 mice. CA1: $N$ = 5 mice, $N$ = 6 mice. (G) Within-session stability of spatially modulated cells in each region active (LS: $n$ = 475 cells, CA1: $n$ = 336 cells, CA3: $n$ = 143 cells). (H) MI per binarized event for all cells (LS: $n$ = 1,030 cells, $n$ = 15 mice. CA1: $n$ = 1251 cells, $n$ = 5 mice, CA3: $n$ = 464 cells, $n$ = 6 mice). *, $p < 0.05$, **, $p < 0.01$, ****, $p < 0.0001$. Test used in F–H: Kruskal–Wallis with Dunn's multiple comparisons test. The underlying data can be found in S1 Data. eGFP, enhanced green fluorescent protein; LS, lateral septum; LSd, dorsal lateral septum; LSi, intermediate lateral septum; MI, mutual information; SFP, spatial footprint.

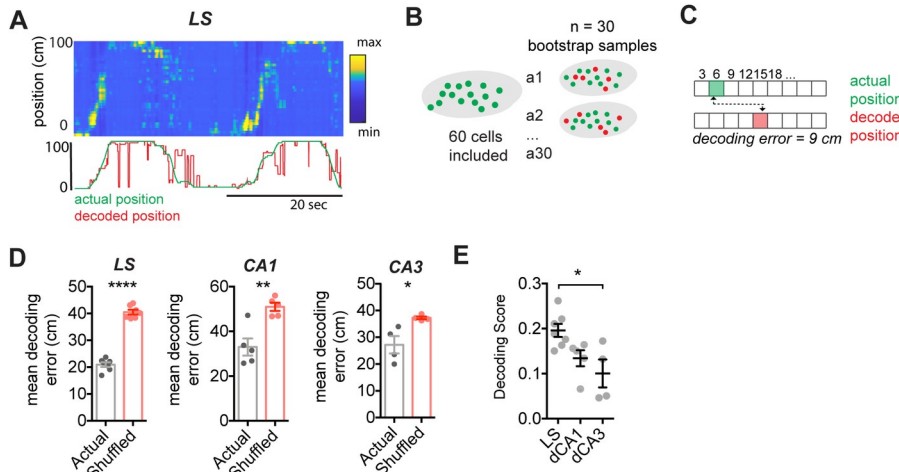

**Fig 2. Location can be reliably decoded from LS GABAergic cells.** (A) Top: posterior probabilities for each frame estimated from binarized calcium activity; bottom: corresponding actual (green) and decoded (red) location. (B) Schematic representation of bootstrapping approach. (C) Method for computing mean Euclidean distance decoding error (in cm) for each bootstrapped estimate. (D) Decoding error for actual vs shuffled dataset (30 bootstrapped samples of 60 cells, LS: $N$ = 7 mice, CA1: $N$ = 5 mice, CA3: LS = 4 mice). (E) Mean decoding score for each region. *, $p < 0.05$, **, $p < 0.01$, ****, $p < 0.0001$. Test used in D, paired $t$ test, E, one-way ANOVA, Holm–Sidak's multiple comparisons test. The underlying data can be found in S1 Data. LS, lateral septum.

Previous studies have used decoding methods to predict behavioral variables from calcium imaging data recorded in the hippocampus [34,38,55], yielding insights into the amount of information encoded by a neuronal assembly. Here, we asked whether we could reliably estimate the mouse location solely from LS neuronal activity patterns (Fig 2). Using 30 bootstrap samples of 60 cells (Fig 2B and 2C), decoding location using LS neuronal activity significantly outperformed a decoder that was trained using shuffled data (paired $t$ test, t(6) = 13.56, $p < 0.0001$; Fig 2D), even when decreasing the number of neurons used (S8A and S8B Fig). Similarly, a decoder trained using CA1 or CA3 data also yielded significantly less error than one using shuffled surrogates (CA1: t(4) = 7.558, $p = 0.0016$; CA3: t(3) = 3.233, $p = 0.0481$; Figs 2D and S8C and S8D). While using a small bootstrap sample size allowed a fair comparison between recording regions, it also induced higher error rates. To ensure the proper functioning of our decoder, we confirmed that the decoding error decreased significantly when increasing bootstrapped sample size (S8C and S8D Fig).

In order to compare decoding accuracy for LS, CA1, and CA3, a decoding score was computed (Fig 2E; see Methods), and we found that LS significantly outperformed CA3 (one-way ANOVA, F(2,13) = 6.277, $p = 0.0124$; Holm–Sidak's multiple comparisons test, $p < 0.05$), but not CA1. Overall, spatially modulated cells recorded in LS were significantly more active than those recorded in the hippocampus (H(3) = 155.1, $p < 0.0001$; S5A and S5B Fig), which could be a contributing factor to the overall higher decoding accuracy using LS cells. To confirm that the differences observed in decoding error are not due to the inclusion of a large number of pseudo-silent cells in the dCA1 or dCA3 bootstrapped samples, we confirmed our findings using an activity cutoff of P(A) > 0.001 (S8E and S8F Fig). Similarly, to confirm that the observed differences in decoding error are not due to the use of a temporal smoothing window, we replicated our results using a decoder that omitted temporal filtering (S8G and S8H Fig).

## LS GABAergic neurons exhibit stable spatial activity during free exploration in a 2D environment

In order to assess spatial coding during free exploration in a 2D environment, we recorded $Ca^{2+}$ activity of 1,899 GABAergic neurons in 28 mice implanted in distinct subregions of the LS, while the animals were freely exploring a novel open field. Out of 1,899 recorded cells, 37.80% (718/1,899 cells, $n$ = 28 mice) of LS cells displayed significant spatial information ($p < 0.01$; Figs 3A and 3B and S9), as compared to 31.33% (323/1,031 cells, $n$ = 6 mice) in CA1 and 25.84% (138/534 cells, $n$ = 8 mice) CA3 cells. Despite the LS' known involvement in feeding-related behaviors [11–14], we did not observe any increased number of cells with firing fields around food zones, nor did we observe overrepresentation of objects or walls (S10 Fig). To assess reward-modulation in the context of increased task demand, we assessed spatial firing of LS GABAergic neurons recorded while animals performed a delayed nonmatching to place task in a T-maze but did not observe any significant reward code compared to shuffled surrogates (S11 Fig).

With theta-rhythmic cells in the LS having higher firing rates than theta-rhythm-independent cells [56], and the suggestion that theta-rhythmic cells could receive direct hippocampal inputs (though see [57,58]), we asked whether LS cells with higher firing rates were also more likely to be spatially modulated. Indeed, spatially modulated cells were significantly more active than nonmodulated cells (Mann–Whitney test, U = 342,219, $p < 0.0001$; S9G and S9H Fig) and displayed a higher bursting index, defined as the probability of a cell being active, given it was already in an active state, or $P(A_t|A_{t-1})$, than nonspatially modulated cells (Mann–Whitney test, U = 368,276, $p < 0.0001$; S9H Fig).

We compared the place field properties of cells recorded in LS to those of dorsal CA1 and CA3 (Fig 3). For each animal, we computed the portion of significantly positionally modulated cells per region, which was not significantly different across LS (36.71 ± 2.16%), dCA1 (38.32 ± 9.36%), and dCA3 (28.39 ± 6.73%; one-way ANOVA, F(2,39) = 1.105, $p$ = 0.3414; Fig 3B). We computed the information for each binarized event and observed that the LS ($1.020 \times 10^4 \pm 7.56 \times 10^7$ bits/binarized event) carries significantly less information per binarized event than CA1 ($1.200 \times 10^4 \pm 1.067 \times 10^6$ bits/binarized event; Kruskal–Wallis, H(3) = 264.6, $p < 0.0001$; Dunn's multiple comparisons test, $p < 0.0001$) and CA3 ($1.199 \times 10^4 \pm 1.605 \times 10^6$ bits/binarized event; $p < 0.0001$). As expected on the basis of prior results [4,28,30,31], within-session stability of LS cells (0.31 ± 0.011) was significantly lower than that of spatially modulated cells recorded in CA1 (0.44 ± 0.017; Kruskal–Wallis, H(3) = 89.52, $p < 0.0001$; Dunn's multiple comparisons test, $p < 0.0001$) and CA3 (0.56 ± 0.027; $p < 0.0001$; Fig 3D), and place fields were more dispersed (LS: 14.47 ± 0.13 cm; CA1: 12.00 ± 0.207 cm; CA3: 10.61 ± 0.35 cm; Kruskal–Wallis, H(3) = 155.7, $p < 0.0001$; Fig 3E). Similar to recordings in a 1D environment, we observed large differences in P(A) between spatially modulated cells recorded from LS and those from dorsal CA1 and CA3 (S12A Fig). Therefore, we also assessed for each region the proportion of spatially modulated cells, split-half stability and mean-dispersion using an activity cutoff of P(A) > 0.001, and confirmed our results (S12B Fig).

When decoding the location of the animals using bootstrap samples of 80 cells from each region (Fig 3F), decoding error was significantly lower in LS (20.16 ± 1.48 cm) compared to shuffled surrogates (26.03 ± 0.84 cm; two-way RM ANOVA, F(1,13) = 89.09, $p < 0.0001$ for main effect of shuffle), even when decreasing the number of neurons used to only 40 neurons (S13A and S13B Fig). Omission of temporal filtering recapitulated these results (S13E Fig). The same held true for decoders trained using CA1 or CA3 data, respectively (S13C and S13D Fig). Strikingly, a decoder trained on data from LS does not perform significantly worse than a decoder trained on CA1 or CA3 data (F(2,13) = 2.186, $p$ = 0.2384 for main effect of region;

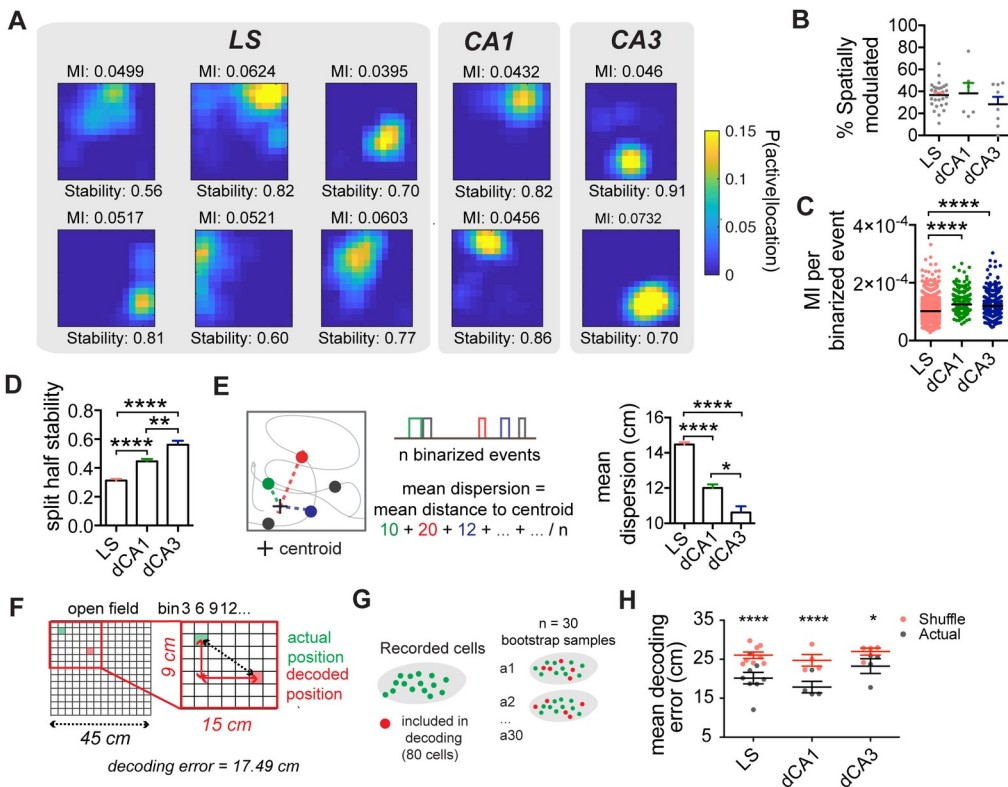

**Fig 3. LS shares spatial coding characteristics with its main hippocampal inputs CA1 and CA3 during free exploration in a 2D environment.** (A) Example tuning maps of spatially modulated cells recorded from LS, dorsal CA1, and dorsal CA3 in a 45 × 45 cm open field (3 × 3 cm bins). (B) Proportion of spatial cells per animal (LS: $n$ = 28 mice; dCA1: $n$ = 6 mice; dCA3: $n$ = 8 mice). (C) MI (bits) per binarized event for all cells recorded from each region (LS: $n$ = 1,899 cells from $n$ = 28 mice; dCA1: $n$ = 1,031 cells, $n$ = 6 mice; dCA3: $n$ = 534 cells, $n$ = 8 mice). (D) Within-session stability for spatial cells (LS: $n$ = 734 spatial cells from $n$ = 28 mice; dCA1: $n$ = 424 spatial cells, $n$ = 6 mice; dCA3: $n$ = 138 spatial cells, $n$ = 7 mice). (E) Left: mean dispersion computation. Right: mean dispersion for all spatial cells recorded from each region. (F) Method for computing the mean decoding error. (G) Bootstrapping approach using 80 randomly selected cells, for 30 bootstrapped samples. (H) Mean decoding error for LS ($n$ = 8 mice), CA1 ($n$ = 4 mice), and CA3 ($n$ = 4 mice). \*1, $p < 0.05$, \*\*, $p < 0.01$, \*\*\*\*, $p < 0.0001$. Test used in B, one-way ANOVA, C–E, Kruskal–Wallis, Dunn's multiple comparison test, H, two-way ANOVA, Sidak's multiple comparisons test. The underlying data can be found in S1 Data. LS, lateral septum; MI, mutual information.

Fig 3H). Similarly to recordings on the linear track, we used an activity cutoff of P(A) > 0.001 to confirm that the differences observed in decoding error between regions are not due to the inclusion of a large number of pseudo-silent cells in the bootstrapped sample (S13F Fig).

## A subset of LS cells has stable spatial representations up to 8 days, similar to CA1

To test whether the LS and the hippocampus exhibit comparable evolutions of their spatial code over time, we recorded from animals implanted in LS and dorsal CA1 in a novel open field over both short (3 days) and longer timescales (8 days; Figs 4 and S14). We assessed the stability of the spatial map across sessions and observed a subset of LS spatially modulated cells that were stable over time (Fig 4B). Correlating the tuning maps of aligned LS cell pairs over days lead to a significant increase in mean pairwise correlation value over days (two-way ANOVA, $F_{(2,808)}$ = 3.259, $p$ = 0.0389, main effect of time) with LS cells being statistically more stable than shuffled comparisons ($F_{(2,808)}$ = 277.0, $p < 0.0001$, main effect of shuffling;

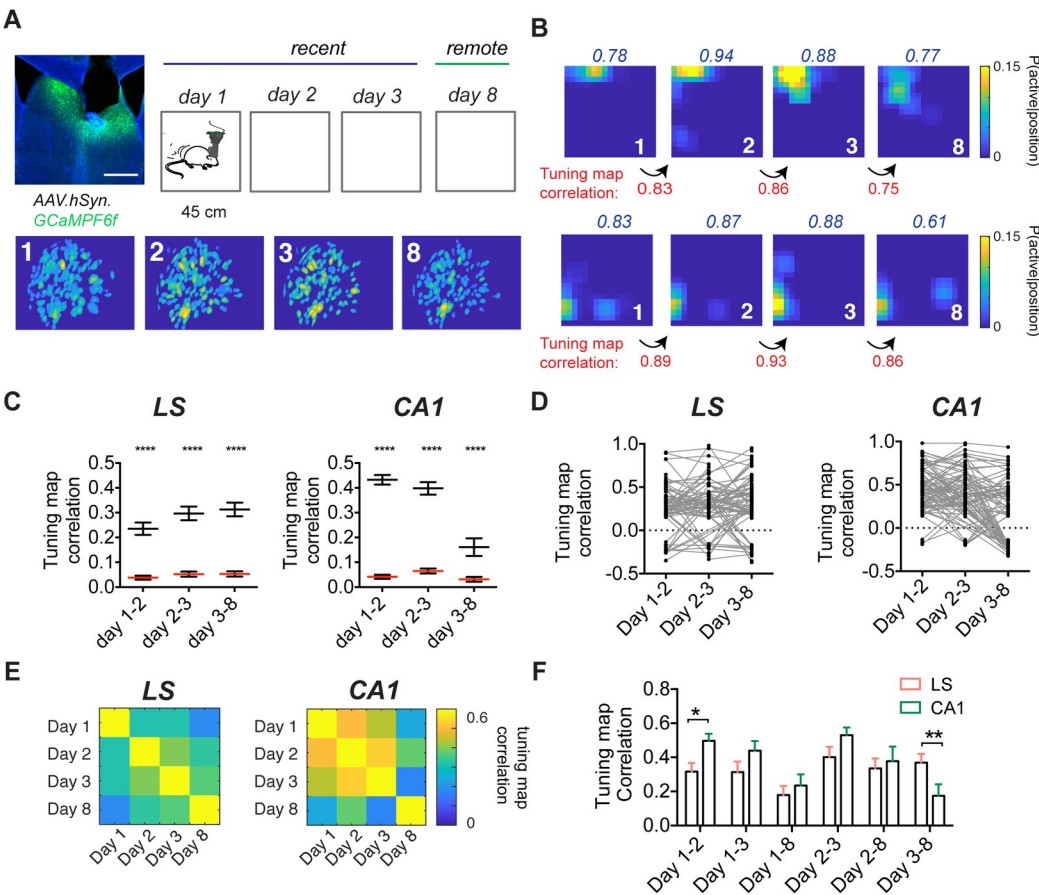

**Fig 4. LS place code remains similar to CA1 over longer periods of time.** (A) Experimental setup (top) with a representative example of an animal implanted in the LS and aligned spatial footprints of cells recorded over days (bottom). (B) Tuning maps for 2 sets of stable cells recorded over all days, with each row being one aligned cell. Within-session correlation indicated in blue. Tuning map correlation indicated at the bottom in red. (C) Significant tuning map correlation for aligned cell pairs (black) vs. shuffled pairs (red) for progressive days for LS (day 1–2, *n* = 157 cells; day 2–3, *n* = 122 cells; day 3–8, *n* = 129 cells; *n* = 5 mice) and dorsal CA1 (day 1–2, *n* = 165 cells; day 2–3, *n* = 122 cells; day 3–8, *n* = 99 cells; *n* = 3 mice). (D) Significant tuning map correlations for all cells found on all days for LS (*n* = 84) and CA1 (*n* = 86). (E) Matrix of mean tuning map correlation for all aligned cells that were significantly spatially modulated on each day, for LS (*n* = 23–37 cells, *n* = 5 mice) and CA1 (*n* = 23–41 cells, *n* = 3 mice). (F) Mean tuning map correlation for data shown in E. *, *p* < 0.05, **, *p* < 0.01, ****, *p* < 0.0001. Test used in C, two-way ANOVA, with Sidak's multiple comparisons test, F, two-way ANOVA, with Fisher's LSD post hoc test. The underlying data can be found in S1 Data. LS, lateral septum.

Figs 4C and S14C). We observed the opposite pattern for pyramidal cells recorded from dorsal CA1, with a significant decrease in mean pairwise correlation value over days (two-way ANOVA, F(2,766) = 33.47, *p* < 0.0001 main effect of time, F(1,766) = 319.8, *p* < 0.0001 main effect of shuffle; Fig 4C). For cells recorded in the LS, we observed the strongest within-session stability on day 3 (Kruskal–Wallis, H(4) = 19.16, *p* = 0.0003; S14E Fig), as well as an increase in spatial information (Kruskal–Wallis, H(4) = 13.27, *p* = 0.0041; S14B Fig). There was no significant increase in the proportion of stable cell pairs over subsequent days (one-way ANOVA, F(3) = 0.400, *p* = 0.9537; S14D Fig), or the proportion of spatially modulated cells per animal for each days (one-way ANOVA, F(3,16) = 1.618, *p* = 0.2247; S14E Fig).

We next assessed the significant tuning map correlations on subsequent day pairs for those cells that were significantly spatially modulated on day 1 and were subsequently found on all recording days (Fig 4D), as well as those that passed our criteria for spatial modulation on all

days (Fig 4E). We then compared the place field correlation for different day pairs ($F(5,363)$ = 2.625, $p$ = 0.0239, interaction effect), and we found that on shorter timescales, tuning maps for LS cells (0.316 ± 0.0507) are significantly less correlated than CA1 (0.497 ± 0.0411; Fisher's LSD day 1 to 2, $p$ = 0.0153; Fig 4F). On the other hand, the correlation for day pairs 3 to 8 is higher for the LS (0.369 ± 0.0513) compared to dorsal CA1 (0.174 ± 0.0669; Fisher's LSD, $p$ = 0.0098; Fig 4F). Together, this suggests that a subset of LS spatial cells encodes spatial information over longer periods of time.

## LS represents direction, velocity, and acceleration information

In CA1, place cells have been found to carry directional [59–61] and speed-related information [61–64]. Previous studies found that subsets of LS neurons show some degree of modulation by direction of travel [31] as well as velocity and acceleration [20] through correlative measures in spatial alternation or T-maze navigation tasks, where the relationship between cell activity patterns and movement-associated variables may be confounded with task-dependent variables. Here, we again employed an information-theoretic approach to compute the MI between calcium activity and each of these self-motion correlates in a subset of mice recorded during a 15-minute free foraging task in the open field. We found that 28.13% of LS cells are significantly modulated by head-direction (346/1,230 cells, $n$ = 19 mice; Figs 5A and S15), 18.70% by velocity (230/1,230 cells; Fig 5C) and 24.63% by acceleration (303/1,230 cells; Fig 5E). We assessed the stability of LS directional and self-motion tuning over short (3 days) and longer timescales (8 days; S16 Fig), and we observed a subset of LS directionally modulated cells that were stable over time (S16A Fig). Mean tuning vectors correlation of aligned LS cell pairs significantly increased over days (two-way ANOVA, $F(2,774)$ = 3.682, $p$ = 0.0256, main effect of time) with LS cells being statistically more stable than shuffled surrogates ($F(1,774)$ = 54,42, $p$ < 0.0001, main effect of shuffling; S16C Fig). For velocity or acceleration tuning, we did not observe such stable tuning over time (two-way ANOVA, velocity: $F(1,506)$ = 2.609, $p$ = 0.1069, acceleration: $F(1,320)$ = 0.1986, $p$ = 0.6561, main effect of shuffle; S16D and S16E Fig).

We compared the proportion of LS cells significantly modulated by each of these variables with the proportion of those cell types found in dCA1 and dCA3. The proportion of directionally modulated cells was significantly different across regions (LS: 27.89 ± 2.16, dCA1: 46.11 ± 4.98, dCA3: 28.92 ± 4.52; Kruskal–Wallis, $H(3)$ = 6.586, $p$ = 0.0371, Fig 5B) with a higher proportion of directionally modulated cells in CA1 as compared to LS (Dunn's multiple comparisons test, $p$ = 0.0340; Fig 5B). The proportion of velocity encoding cells was not significantly different between LS (20.43 ± 2.598), CA1 (24.38 ± 7.73), or CA3 (20.04 ± 3.62; Kruskal–Wallis, $H(3)$ = 0.1016, $p$ = 0.9504, Fig 5D). Similarly, no difference was found in the proportion of acceleration encoding cells between LS (25.11 ± 2.34), CA1 (24.14 ± 4.13), or CA3 (22.60 ± 2.98, Kruskal–Wallis, $H(3)$ = 0.3337, $p$ = 0.8463; Fig 5F). We found a proportion of cells that were significantly tuned to more than one of these variables in both LS as well as dCA1 and dCA3 (Figs 5G, S15A–S15D and S17), with 22.03% of LS cells (271/1,230 cells) encoding more than one modality (Fig 5H). Together, this conjunctive coding for location, directionality, and velocity indicates that LS cells fire in response to more complex navigational features, similarly to the hippocampus.

To understand the function of this downstream copy of the hippocampal spatial code, we assessed the downstream targets of the dCA1/dCA3 to LS projection. First, we used a Cre-dependent, anterograde tracing approach to confirm LS projections to the hypothalamus and ventral tegmental area (VTA) in the mouse (S18 Fig), as it was initially described in the rat [3,5,25]. Next, we leveraged the anterograde transsynaptic properties of Cre-expressing AAV1

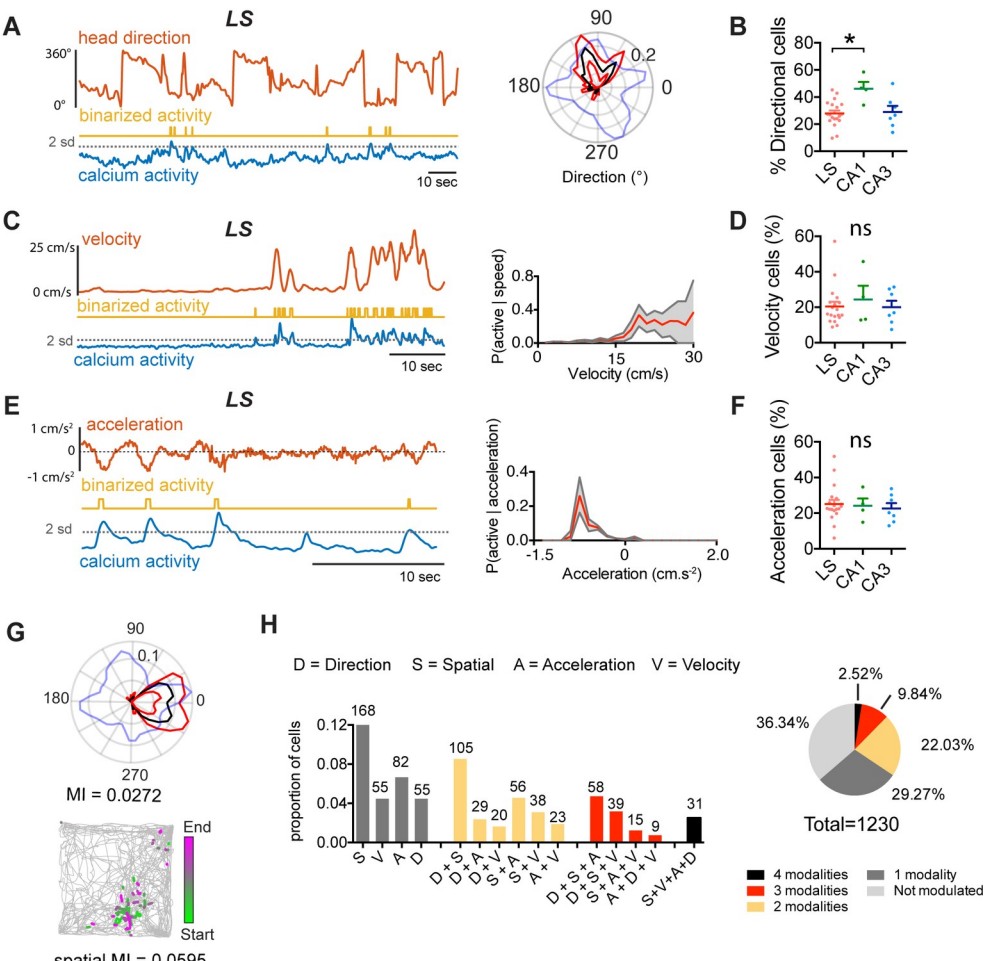

**Fig 5. LS GABAergic cells encode direction, speed, and acceleration.** (A) Left: activity of an example LS neuron during free exploration in an open field; top: HD (red); middle: binarized activity (yellow); bottom: raw calcium activity (blue). Right: corresponding polar plot indicating the probability of the cell being active as a function of the animals' HD with (black, p(active | direction); red lines indicate 95% upper and lower percentile. Blue line indicates the normalized time spent for each direction. MI calculated using 40 bins of 9° (S15 Fig). (B) Proportion of significantly directionally modulated cells in LS, dCA1, and dCA3. (C) Left: activity of an example LS neuron during free exploration modulated by velocity (top, red); middle: binarized activity (yellow); bottom: raw calcium activity (blue). Right: Tuning curve for velocity for the example cell (red) with 95% upper and lower percentile in gray. MI calculated using 20 bins. (D) Proportion of significantly speed modulated cells in LS, dCA1, and dCA3. (E) Left: activity of an example LS neuron during free exploration modulated by acceleration (top, red); middle: binarized activity (yellow); bottom: raw calcium activity (blue). Right: Tuning curve for acceleration for the example cell (red) with 95% upper and lower percentile in gray. MI calculated using 20 bins. (F) Comparison of proportions of significantly modulated cells for each region. (G) Example of an LS cell that is both significantly head direction modulated (left), as well as spatially modulated (right). (H) Left: proportion of cells that are significantly modulated by only one modality (gray), 2 modalities (yellow), 3 (red) or all 4 of the investigated variables (black). The number above the bars indicates the absolute number of cells found to be modulated in the total population ($n = 1,230$ cells, $n = 19$ mice). Right: absolute proportion of cells modulated by any combination of variables. Test used in B, D, F: Kruskal–Wallis test with Dunn's multiple comparisons test. *, $p < 0.05$. The underlying data can be found in S1 Data. A, acceleration; D, direction; HD, head direction; LS, lateral septum; MI, mutual information; ns, not significant; S, spatial coding; V, velocity.

viral vector [65] and assessed the targets of the dorsal CA1-lateral septum (dCA1-LS) as well as dorsal CA3-LS (dCA3-LS) projections (Fig 6). For either dCA1 or dCA3 injections (Figs 6A and S19), we observed a direct pathway from the principal cells of the hippocampus, via the LS, leading to dense innervation of the MS (Fig 6B) and the hypothalamic regions (Fig 6C).

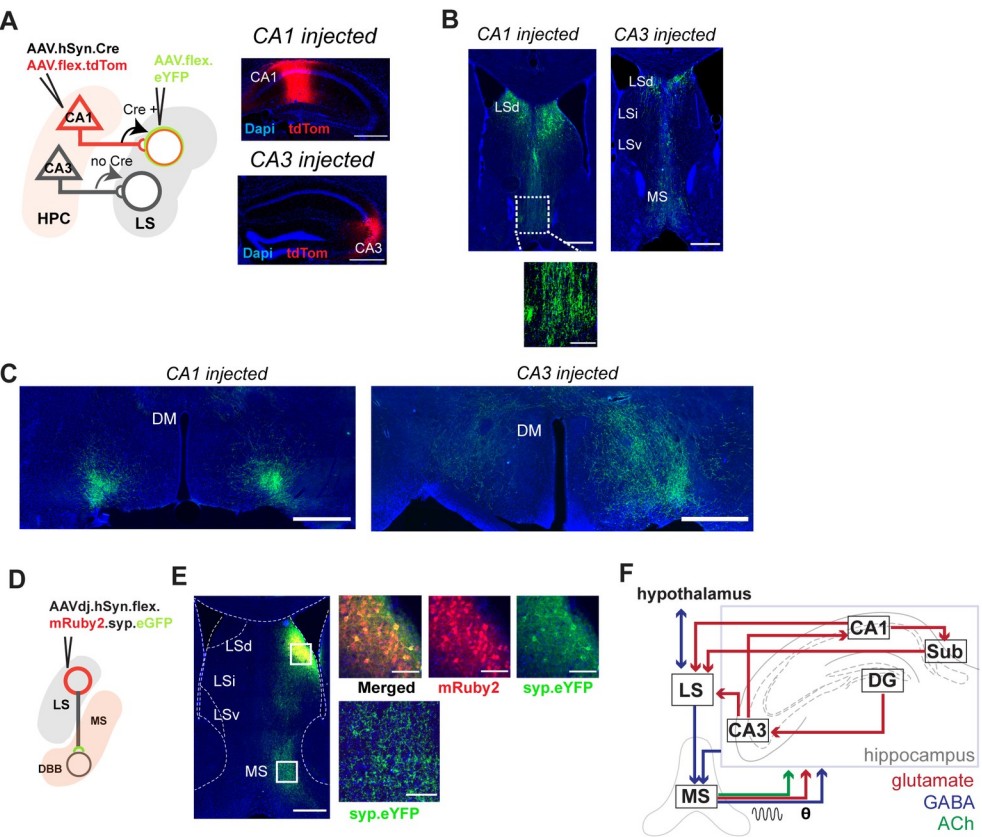

**Fig 6. LS cells receiving hippocampal inputs project directly to hypothalamus and MS.** (A) Diagram of dual-viral injection strategy for anterograde transsynaptic tracing. The diagram is based on dorsal CA1 targeted injection; dCA3 is also used for injections, together with coronal sections showing primary injection sites in dorsal CA1 (top) or dorsal CA3 (bottom). Red: tdTom expression; blue: DAPI, with zoomed images showing tdTom-positive cell bodies are predominantly located in the pyramidal layer (bottom). For additional images of spread of injection, see S19 Fig. (B) eYFP-positive cell bodies at the anterior dorsal LS and fibers at the level of the MS following dCA1 injection (left) and dCA3 injection (right). Inset left: eYFP-positive fibers at the level of MS. (C) eYFP-positive axons are seen bilaterally at the level of the LH following dCA1 injection (left) and dCA3 injection (right). (D) Injection strategy for Cre-dependent AAV. Synaptag mediated tracing in the LS. (E) Coronal section of dorsal LS, with synaptophysin-bound eYFP at the MS. Top right: zoomed images showing transduction at the injection site. Bottom right: eYFP-positive fibers. (F) Schematic with proposed connections of the LS within the hippocampal network. Scale bars: B, left and right: 800 μm. C, left and right: 500 μm. The underlying data can be found in S1 Data. DBB, diagonal band of Broca; DG, dentate gyrus; DM, dorsomedial hypothalamic nucleus; HPC, hippocampus; LH, lateral hypothalamus; LS, lateral septum; LSd, dorsal lateral septum; LSi, intermediate lateral septum; LSv, ventral lateral septum; MS, medial septum.

We observed that dCA1-LS preferentially targets the LH, and dCA3-LS targets the hypothalamus more broadly as well as the nuclei of the medial zone. Whether the GABAergic cells of the LS form functional synapses at the level of the MS/diagonal band complex is debated [5,26,66–68]. Thus, in order to determine whether the projections observed at the level of the MS with both our anterograde tracing approaches constituted passing fibers or synaptic connections, we used an AAV-Flex-Synaptag [69], a Cre-dependent viral construct to express mRuby in the cytosol and eGFP bound to synaptophysin, a protein primarily expressed in the presynaptic endings of cells participating in synaptic transmission (Fig 6D). For this, we used a CaMKIIα-Cre mouse line, expressing Cre only in CAMKII-positive cells, a marker that is abundant in the GABAergic cells of the LS but absent in the MS, thereby preventing any unspecific labeling. After expressing AAV-Flex-Synaptag in the intermediate LS, we observed significant synaptophysin-bound projections in the MS (Fig 6E), confirming the existence of a synaptic interface

between the LS and the MS. Together, our work shows that LS receives direct inputs from dorsal CA1 and CA3 and, in turn, projects to regions that are not directly receiving inputs from the hippocampus itself (Fig 6F).

## LS cells coding for space, direction, and self-motion are nonuniformly distributed

To map out the monosynaptic connections between hippocampal and LS neurons, we injected the anterograde transsynaptic Cre-expressing AAV1 viral vector [65], in either dCA1 or dCA3 (S19 and S20A Figs). We subsequently injected a Cre-dependent eYFP expressing virus in the LS to visualize cells that receive direct hippocampal input (S20B Fig). Importantly, this approach allowed us to quantify the cells along the LS anterior–posterior and dorsal–ventral axis receiving direct hippocampal inputs and assess their distribution density within the anterior regions of the LS (Bregma +1.0 to 0.0 anterior–posterior). Because we can only observe secondary labeled cell bodies at locations where there was infusion of the secondary, Cre-dependent virus, we used a relatively large injection volume (total 800 nL, targeted at the intermediate LS). Both dorsal dCA1 and dCA3 injections resulted in a majority of labeled LS cell bodies in dorsal regions compared to intermediate and ventral LS (S20D–S20F Fig). Hence, LS neurons receiving direct inputs from dorsal CA1 and CA3 appear to be localized at more dorsal–posterior regions as compared to more ventral–anterior regions of LS.

Next, we examined whether spatially modulated cells recorded in LS had spatial properties that were dependent on the localization of their recording. To measure this, we classified the location of gradient refractive index (GRIN) lens implants along the dorsal–ventral and anterior–posterior axis (Fig 7A). Postmortem histological verification of GRIN lens implant combined with within-animal analysis of the location of recorded cells enabled us to approximate the location of each cell (Fig 7B). While we observed no differences in cell activity characteristics, such as bursting index or activity probability along the anterior–posterior axis, medial–lateral axis, or dorsal–ventral axis (S21 Fig), we found a pronounced increase in the portion of stable spatially modulated cells (within-session stability > 0.5) along the anterior–posterior axis (linear regression, $R^2 = 0.1071$, $p = 0.0022$, the proportion of stable cells per 0.2 mm bin per animal, $n = 24$ mice; Fig 7C). We observed a similar gradient along the dorsal–ventral axis, with a larger proportion of stable spatial cells found at the more dorsal regions of the LS, although this trend failed to reach significance (linear regression, $R^2 = 0.1399$, $p = 0.0718$, $n = 24$ mice; Fig 7C). Taken together, these data strongly suggest that regions of the LS that receive stronger innervation from the dorsal hippocampus have a larger proportion of cells that reliably encode space. This suggests that this information could be directly inherited from the hippocampus, which should be tested using targeted inactivation of the hippocampal pyramidal inputs to LS.

We also assessed the topographical distribution of direction and self-motion tuned cells within the LS. We observed a relatively strong dorsal–ventral gradient in the distribution of stable (within-session stability > 0.5) directionally modulated cells (linear regression, $R^2 = 0.3837$, $p = 0.0061$, $n = 17$ mice; Fig 7C), with most directionally modulated cells located at the more dorsal regions of the LS, but we did not observe such a gradient along the anterior–posterior axis (linear regression, $R^2 = 0.0544$, $p = 0.0615$, $n = 24$ mice; Fig 7C). Velocity-modulated cells were primarily found at the ventral pole of the LS (linear regression, $R^2 = 0.1836$, $p = 0.0367$, $n = 24$ mice), but no such gradient was found along the anterior–posterior axis (linear regression, $R^2 = 0.00348$ $p = 0.0874$, $n = 24$ mice). Acceleration-modulated cells were found evenly distributed throughout the LS (DV: $R^2 = 0.0133$, $p = 0.2928$; AP: $R^2 = 0.01332$, $p = 0.2929$, $n = 24$ mice).

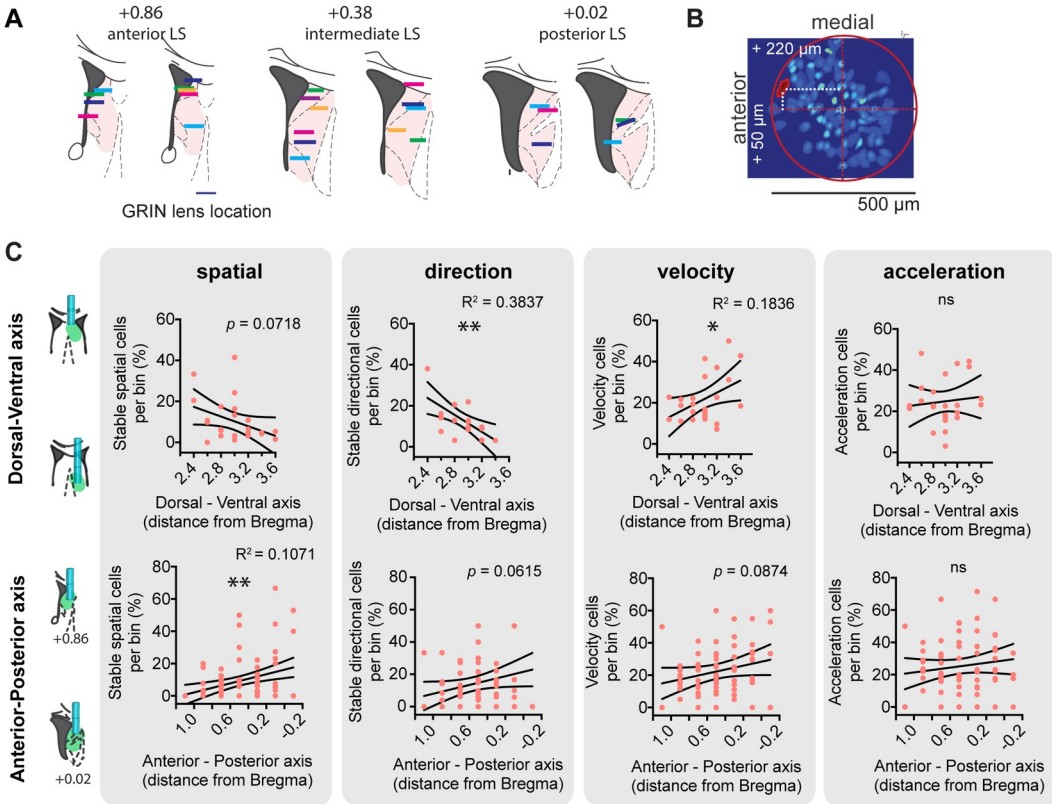

**Fig 7. LS cells coding for space, direction, and self-motion correlates are nonuniformly distributed along the dorsal–ventral and anterior–posterior axis.** (A) Strategy to record from different anterior–posterior levels in LS (left) and implantation sites covered. (B) Strategy to approximate cell location. Background: maximal projection of representative recording; red: outline of approximated GRIN lens position; white: distance from the center of the GRIN lens to the cells of interest. (C) Top: for 0.2 mm bins along dorsal–ventral axis, proportion of stable (within-session stability > 0.5) spatially modulated cells ($n$ = 24 mice), directionally modulated cells ($n$ = 17 mice), velocity cells ($n$ = 24 mice), and acceleration cells ($n$ = 24 mice), respectively. Bottom: same as top for anterior–posterior axis. Each red dot represents the average per animal per bin along the anterior–posterior axis (left) and along the dorsal–ventral axis. Black lines indicate 95% confidence intervals. *, $p < 0.05$, **, $p < 0.01$. Test used in C: linear regression. The underlying data can be found in S1 Data. GRIN, gradient refractive index; LS, lateral septum; ns, not significant.

## Discussion

While spatial coding has been extensively characterized in the hippocampal formation, how downstream regions integrate this information has only recently begun to receive attention. The LS has reemerged as a critical region implicated in a space encoding network, although electrophysiological recordings of the LS in rats have led to disparate estimates of the quantity and quality of place cells in the region [4,20,28,32]: LS place cells recorded on a circular track were described to be almost absent [4] but were found to be abundant in reward-seeking tasks [20,32]. Using a large calcium imaging dataset and unbiased information metric and decoding approaches, we found that 37.80% of GABAergic LS cells robustly code for space during free exploration in an open field and 43.90% of cells during linear track alternation. One of the major features of the linear track is the induction of directional place fields in LS, which may underlie some of the differences we observed in 1D versus 2D spatial coding characteristics.

In addition to spatial information, we found that the LS also reliably encodes velocity, acceleration, and directional information, suggesting that the LS encodes more complex navigational features than previously thought. Neuronal representations in the hippocampus have

been found to change over time, with place code ensembles changing rapidly: An estimated approximately 40% [34] to approximately 75% to 85% [38] of CA1 neurons were found to remap over days. The stability of the LS spatial code has received little attention to date. In the current study, we found a subpopulation of cells in the posterior LS that display stable place fields over 8 days. The functional role of such stability remains to be elucidated. One possibility is that this ensemble may mediate the stable encoding of contexts over longer durations of time, which could account for the critical importance of the LS for a number of spatial behaviors, including contextual fear conditioning [16,41–44], context-reward associations [39,40], and radial-maze navigation tasks [45,46].

Similar to the change in properties of hippocampal place cells along the dorsal–ventral axis [70,71], a previous work has hinted at a dorsal to ventral organization of the LS place code: Electrophysiology studies recorded a slightly larger proportion of LS place cells in the more dorsal regions [30]. A complementary anatomical microstructure has been previously described for the rate-independent phase code in the LS, where the strength of the phase code increased as a function of recording depth along the dorsal–ventral axis [4]. Here, we find that lateral septal spatially modulated cells are arranged along an anterior–posterior gradient similar to the gradient of hippocampal inputs into the LS—a finding that helps to reconcile the variation in previously reported estimates of spatial cells in the LS, which ranged between 5.3% of cells recorded along the dorsal–ventral axis [4] up to 56% recorded at the most dorsal level of the LS [20]. This pattern suggests that the inhibitory neurons of the LS may inherit a place code from the hippocampus, with subregions receiving dorsal hippocampal inputs being most similar to the classical hippocampal place cells. One potential caveat of using miniaturized microscope when comparing hippocampus and LS coding properties is that of circuit damage due to GRIN lens implantation, which may be particularly the case for CA3 implants due to the occasional damage to CA1. Interestingly, the LS place code described here is not the only case of GABAergic neurons inheriting some spatial properties from hippocampal pyramidal cells: Interneurons within CA1 have been shown to exhibit significant spatial modulation and comparable information as pyramidal place cells, although with a higher firing rate and greater spatial dispersion [72]. It should be considered that, although the hippocampus seems a likely source of the positional information encoded by the LS, this information could also arise from incoming projections from any other region projecting to LS, such as the entorhinal cortex [5,25], which could account for part of head direction and velocity tuning.

In addition to spatial modulation of LS cells, we observed a significant number of velocity- and acceleration-modulated cells in the LS. These cells have been previously described by others on the basis of correlative measures, although estimated proportions vary widely: For velocity tuning in reward-seeking tasks, estimates range from almost absent [4] to almost 60% of cells [20]. For acceleration, approximately 45% of LS cells were found to show some degree of correlation, with almost 30% of recorded cells correlated with both speed and acceleration [20]. Here, using information metrics and a free exploration task in which firing characteristics are not confounded with task parameters, we find that 18.70% of cells were modulated by velocity and 24.63% by acceleration. Cells reliably coding for both velocity and acceleration were relatively rare (2%). Velocity and acceleration cells were distributed dissimilarly in the LS, with velocity cells being more abundant toward the ventral pole, whereas acceleration-modulated cells were found to be distributed equally. This raises the question whether velocity and acceleration information are inherited from different areas upstream of the LS. Interestingly, cells coding for direction were more abundant toward the dorsal portion of the LS but distributed equally along the anterior–posterior axis. Previous studies have identified classical head direction cells in multiple, well-described circuits, including the postsubiculum [73,74], anterodorsal thalamic nucleus [75], mammillary bodies [76], entorhinal cortex [77],

retrosplenial cortex [78,79], and parasubiculum [80,81]. A likely source of directional information to the LS is the entorhinal cortex [82]. Our work supports the perspective that LS neurons combine a wide range of modalities and may form a complex representation of context over longer timescales than previously reported. In addition to space, velocity, and reward information, hippocampal pyramidal cells have been shown to encode variables such as time [83,84], odor [85,86], and sound frequency [87]. Whether this information is also relayed to the LS remains to be elucidated. Despite this interconnectedness with the LH and previous reports of LS place cells being skewed toward reward in a spatial navigation task [32], we did not observe any overrepresentation of LS spatially modulated cells around food zones during free exploration or around the reward zone in a T-maze non-match-to-place paradigm.

The question arises of what the function could be of having a region downstream of the hippocampus that expresses such seemingly similar coding characteristics. Previous work estimated the convergence of hippocampal efferents onto LS neurons to be 20 to 800 times denser than to any of its cortical targets [4]. Understanding how individual LS neurons integrate the thousands of synaptic inputs they receive from these hippocampal pyramidal neurons will be critical to understanding how the hippocampal map is processed downstream. Through the process of synaptic integration, a target neuron can fire an action potential upon receiving sufficient temporally coincidental excitatory input on its dendrites. The activation of an LS neuron could thus require multiple hippocampal pyramidal cells to spike simultaneously or in close temporal proximity. Hippocampal pyramidal cells will fire concomitantly when their place fields are in overlapping regions or in close vicinity of another. Due to the high interconnectivity of LS GABAergic neurons [88,89], the activation of one LS cell could subsequently lead to the inhibition of other neurons in the nucleus [24,90], thereby reducing noise. Thus, through this process of coincidence detection and recurrent inhibition, the spatial map could converge to be represented by a much lower number of cells. As such, the LS could accurately convey hippocampal information to downstream regions using a much more distributed code and thereby support more effective information transmission and associative learning [91,92].

Finally, our tracing work shows LS cells receiving hippocampal inputs from principal cells in dorsal CA1 and CA3 in turn project directly to the LH and the MS. Although the existence of an LS–MS projection has long been debated [5,26,66,68,93], we observed both dense innervation at the level of the MS after transduction of dCA1-input and dCA3-input receiving LS cells, as well as synaptic densities between the LS and MS. This suggests that principal cells of the hippocampus send indirect projections to the MS through LS neurons. This circuit is complementary to the well-described projections of hippocampal GABAergic interneurons directly to the MS, which may play a key role in theta-rhythm generation [94,95]. The MS plays a role in generating and propagating theta-rhythms throughout the hippocampal formation [96–98], which, in turn, organizes hippocampal place cell activity [99], as well as a behavioral role in the initiation and velocity of locomotion [100,101]. One hypothesis is that the information on locomotion-related information reported in specific MS neurons [101] may originate from the LS. Moreover, we observed a direct projection from CA1 and CA3, via the LS, to the hypothalamus. This pathway was previously shown by previously using nonspecific monosynaptic anterograde tracing [3], with similar unilateral projection patterns for CA1 and bilaterally for CA3 as described here. Here, we show for the first time that the same LS neurons that receive inputs from dorsal CA1 and CA3 project directly to the hypothalamus. The hypothalamus is a highly connected region known for its role in regulating feeding behaviors [102], arousal and motivation [103], and, more recently, learning and memory [104–107]. Additionally, the LH is crucial for the control of locomotion [108–111] and thought to mediate motivational and goal-directed processes underlying feeding [12,102,112,113].

Together, our findings show that GABAergic cells of the LS may provide the hypothalamus and MS with information about location, direction, and speed, and therefore constitute a core node within a distributed spatial memory network. Within this network, the LS may be necessary for the translation of spatial information to goal-directed or environmentally appropriate actions necessary for survival.

## Methods

### Ethics statement

All procedures were approved by the McGill University and Douglas Hospital Research Centre Animal Use and Care Committee and in accordance with the Canadian Council on Animal Care (protocol 2015–7650).

### Animals

Naive mice (8 to 16 weeks old) were housed individually on a 12-hour light/dark cycle at 22°C and 40% humidity with food and water ad libitum. All experiments were carried out during the light portion of the light/dark cycle. Both male and female mice were used for this study.

### Virus injections

Mice, 8 to 16 weeks old, were anesthetized with isoflurane (5% induction, 0.5% to 2% maintenance) and placed in a stereotaxic frame (David Kopf Instruments). Body temperature was maintained with a heating pad, and eyes were hydrated with gel (Optixcare). Carprofen (10 ml/kg) was administered subcutaneously. All viral injections were performed using a glass pipette connected to a Nanoject III (Drummond) injector at a flow rate of 1 nL/second. All stereotaxic coordinates are taken from Bregma (in mm). After surgery, animals were continuously monitored until recovery. For 3 days after surgery, mice were provided with a soft diet supplemented with Carprofen for pain management (MediGel CPF, approximately 5 mg/kg/day).

**Virus-mediated expression of genetically encoded calcium indicator.**  Adeno-associated virus (AAV) of serotype 9 containing the genetically encoded calcium indicator (GECI) 6 fast (GCaMP6f) under a flex promoter was obtained from the University of Pennsylvania Vector Core (AAV2/9.Syn.Flex.GCaMP6f.WPRE.SV40, CS0641, Penn Vector Core via Addgene). LS viral injections were targeted at the following coordinates: Anterior LS, AP: 0.86; ML: 0.35; DV: −3.0; Intermediate LS, AP: 0.38; ML: 0.35; DV: −2.7, Posterior LS, AP: 0.10; ML: 0.35; DV: −2.5, with DV + or − 0.2 to adjust for more dorsal or ventral placement of injection. LS injections were done in VGAT-IRES-Cre transgenic mice (Slc32a1$^{tm2(cre)Lowl}$/J, JAX stock #016962). For hippocampal principal cell imaging, C57Bl/6J mice (JAX stock #000664) were injected with a viral vector specifically targeting GCaMP6f expression to CaMKII-positive cells (AAV5.CamKII.GCaMP6f.WPRE.SV40, Penn Vector Core via Addgene, CS1024). GCaMP6f injections were targeted at the following coordinates: CA1, AP: −1.8; ML: 1.5; DV: −1.5; CA3, AP: −2.1; ML: 2.3; DV: −2.2. Three of CA3 animals were Grik4-Cre transgenic mice (C57BL/6-Tg(Grik4-cre)G32-4Stl/J, JAX stock #006474), restricting GCaMP6f expression to the pyramidal cells of CA3 specifically.

**Retrograde tracing.**  Retrograde tracing studies were done using a Cre-dependent retrograde Rabies tracing approach. Briefly, 100 nL of helper virus (AAVdj.hSyn.Flex.TVA.P2A.eGFP.2A.oG, NeuroPhotonics Center, Laval, Canada; [114,115]) was injected in the LSd (coordinates as above) of VGAT-IRES-Cre transgenic mice (*n* = 5 animals, Slc32a1$^{tm2(cre)Lowl}$/J, JAX stock #016962). Cre specificity was tested by the NeuroPhotonics center. After 4 weeks of incubation time to allow for the complete expression of the helper virus, 200 nL of inactivated

 

EnvA pseudotyped rabies virus coupled to mCherry was injected (RABV-EnvA-deltaG-mCherry, NeuroPhotonics Center) at the same coordinates as the helper virus injection, and brains were collected 7 days later.

**Anterograde transsynaptic tracing.** To distinguish synaptic targets from passing projection fibers of dCA1 and dCA3 within the LS, C57Bl/6J mice (JAX stock #000664) received a bilateral injection of transsynaptic Cre-inducing anterograde virus (AAV1-hSyn-Cre-WPRE-hGH, Penn Vector Core via Addgene, V24784) mixed with a Cre-dependent tdTom-expressing reporter virus to visualize injection sites (AAV2-Ef1a-flex-tdTomato, NeuroPhotonics Center, AAV602) in a 3:1 ratio, 80 nL total injection volume per hemisphere (*n* = 5 mice per group). Injections were targeted at the following coordinates: CA1: AP: −1.86; ML: 1.5; DV: −1.5; CA3: AP: −2.1; ML: 2.3; DV: −2.2. Following 1 week of incubation time, animals received a bilateral injection of a Cre-dependent anterograde vector in LS expressing enhanced yellow fluorescent protein (400 nL, bilateral, AP: 0.5; ML: 0.35; DV: −2.6, AAVdj-DIO-Cheta-eYFP, NeuroPhotonics Center, AAV938). Brains were collected and processed 3 weeks postinjection.

## Synaptag tracing

To verify presence of synaptic connections at the level of the LS−MS projection, AAVdj-hSyn-flex-mRuby2-syp-eGFP construct (lot AAV799, NeuroPhotonics Center; Oh and colleagues, 2014) was injected in the LS of CaMKIIα-Cre mice (B6.Cg-Tg(Camk2a-cre)T29-1Stl/J, JAX stock #005359), with 100 nL total injection volume injected at 1 nL/second (unilateral, AP: 0.5; ML: 0.35; DV: −2.7). After 4 weeks of incubation time, animals were perfused, and brains were processed as described below.

## GRIN lens implant and baseplate

Two to 4 weeks post-GCaMP6f injection, a 0.5-mm diameter gradient refractive index (GRIN, Inscopix, Palo Alto, California, United States of America) lens was implanted above LS, dorsal CA1 or CA3, or a 1.8-mm GRIN (Edmund Optics, Barrington, New Jersey, United States of America) lens was implanted above dorsal CA1. For a 0.5-mm LS lens implant, the lens was glued to an aluminum baseplate prior to the surgery, which provided 2 main advantages: (1) it allowed us to implant the baseplate/lens assembly in one surgery (instead of separate surgeries); and (2) during the surgery, the miniscope (V3, miniscope.org) was used to image for fluorescence while lowering the lens. The mouse was anesthetized with isoflurane, and the skull was cleared. An approximately 0.6-mm diameter hole was drilled in the skull at the level of the injection site. An anchor screw was placed above the contralateral cerebellum to stabilize the implant. The baseplate and GRIN combination were attached to the miniscope and together secured to the stereotaxic frame. After making a leading track using a 0.5-mm diameter needle, the implant was lowered in place. While lowering the GRIN lens into the tissue, an increase in fluorescent signal indicates that the injection site has been reached. Both the GRIN lens and baseplate were permanently secured using C&B-Metabond (Patterson Dental, Montreal, Quebec, Canada). A plastic cap was used to protect the GRIN lens from scratching during the recovery period and in between recording sessions. The animal was given 4 to 6 weeks of recovery time before experiments started. LS implants were targeted at the following coordinates: Anterior LS, AP: 0.86; ML: 0.35; DV: −2.9; Intermediate LS, AP: 0.38; ML: 0.35; DV: −2.65; Posterior LS, AP: 0.10; ML: 0.35; DV: −2.45; with DV + or − 0.2 to adjust for dorsal, ventral, or intermediate targets; CA3 implants were targeted at AP: −2.1; ML: 2.2; DV: −2.2.

For 1.8-mm GRIN lens implantation, an approximately 2-mm diameter cranial window was prepared. Again, an anchor screw was used to secure the implant in place. After removing the dura, a portion of the cortex above the injection site was gently aspirated using a vacuum

pump without applying pressure on the underlying hippocampal tissue. The 1.8-mm GRIN lens was lowered at the following coordinates: AP: −1.8; ML: 1.5; DV: −1.8. The GRIN lens was permanently attached to the skull using Metabond, and Kwik-Sil (World Precision Instruments, Sarasota, Florida, United States of America) silicone adhesive was placed on the GRIN to protect it. On average 4 weeks postimplant, the silicone cap was removed, and CA1 was imaged using a miniscope mounted with an aluminum base plate while the mouse was under light anesthesia (approximately 0.5% isoflurane) to allow the visualization of cell activity. When a satisfying field of view was found (large number of neurons, visible landmarks), the baseplate was cemented above the GRIN lens, and a protective cap was used to protect the GRIN from scratches. Imaging would start approximately 1 week after baseplating.

## Miniscope recordings

The animals were habituated by being gently handled for approximately 5 minutes per day for 3 days. Animals were then water or chow scheduled (2-hour access per day). Miniscope recordings performed at 30 Hz for 15 minutes, with just one recording session per day to minimize photobleaching. For open field recordings, animals were freely foraging for randomly placed 10% sucrose or 10% sweetened condensed milk in water rewards.

Recording environments consisted of a 45 × 45 cm dark gray open field with visual cues, or a 49 × 49 cm white plexiglass open field, which was placed in different recording chambers. For linear track recordings, rewards (30 μl of 10% sucrose water) were placed at each end of the 100-cm linear track, or a 130-cm linear track, and the mouse had to consume one reward before getting the next one delivered. To ensure that all animals received the same amount of exposure, for analysis of spatial and self-motion modulation, only recordings were included from the first time the animal spent time in the open field.

For analysis of modulation by food zones or objects, animals were placed again in the, now familiar, open field on days 1, 3, and 5. On days 2 and 4, food zones or objects were added in opposing corners of the familiar open field without changing any of the existing visual cues. Food zones or objects were visible to the animal. Time spent in the region of interest was computed by taking all timestamps in which animals had their nose within less than 2 cm from the edge of the food zone or the object.

Nonmatch to place paradigm: Mice were water scheduled (2 hours per day) and trained in an automatized T-maze (MazeEngineers, Skokie, Illinois, United States of America) to a nonmatch to place task. Briefly, each trial was divided into 2 distinct phases: sample and test. In the initial sample phase, a randomly selected arm was blocked, which forced mice to explore the opposite arm where the animal received a 10% sucrose water reward. After consuming the reward, animals would return to the start box for the test phase, during which both arms could be explored. In the test phase, only the previously unexplored arm is baited, so that mice have to alternate locations between sample and test phases. Mice were subjected to 10 trials (sample + test) per day, for 10 consecutive days, and calcium activity in LS was recorded on all days. Data provided in S11 Fig are from day 10 of training.

## Miniscope and behavior video acquisition

Miniscopes (V3 and V4, miniscope.org) were assembled using open-source plans as described previously (miniscope.org) [116,117]. Imaging data were acquired using a CMOS imaging sensor (Aptina, MT9V032) and multiplexed through a lightweight coaxial cable. Data were acquired using a data acquisition (DAQ) box connected via a USB host controller (Cypress, CYUSB3013). Animal behavior was recorded using a webcam mounted above the environment. Calcium and behavioral data were recorded using Miniscope custom acquisition

software. The DAQ simultaneously acquired behavioral and cellular imaging streams at 30 Hz as uncompressed.avi files (1,000 frames per file), and all recorded frames were timestamped in order to perform the subsequent alignment.

## Histology

After completion of all experiments, animals were deeply anesthetized with ketamine/xylazine/acepromazide (100, 16, 3 mg/kg, respectively, intraperitoneal injection). Mice were then transcardially perfused with 4% paraformaldehyde in PBS (PFA). Brains were extracted and postfixed in PFA at 4°C for a minimum of 48 hours. Brains were sectioned at 50 μm using a vibratome and cryoprotected in a solution of 30% ethylene glycol, 30% glycerol, and 40% PBS until used.

## Immunohistochemistry

Sections were washed 3 × 20 minutes in PGT (0.45% Gelatin and 0.25% Triton in 1× PBS) at room temperature. Next, primary antibodies (1:1,000 goat anti-GFP from Novus Biologicals (Littleton, Colorado, United States of America) or 1:10,000 rabbit anti-RFP from VWR (Rockland)) were incubated with PGT overnight at 4°C. Following 10-, 20-, and 40-minute washes in PGT, sections were incubated with secondary antibodies (1:1,000 donkey anti-goat coupled to A488 or 1:1,000 donkey anti-rabbit coupled to A555, both from Molecular Probes (Eugene, Oregon, United States)) in PGT at room temperature for 1 hour. Sections were subsequently washed for 10 and 20 minutes in PGT and 30 minutes in PBS. Sections were then mounted on glass slides and permanently coverslipped with Fluoromount that contained DAPI.

## In vitro patch clamp electrophysiology

Mice were deeply anesthetized, and acute brain slices were obtained following the protective recovery method [118]. Briefly, mice were transcardially perfused with N-methyl-D-glutamine (NMDG-based) solution containing (in mM) 93 NMDG, 93 HCl, 2.5 KCl, 1.2 NaH2PO4, 30 NaHCO3, 20 HEPES, 25 glucose, 5 sodium ascorbate, 2 thiourea, 3 sodium pyruvate, 10 MgSO$_4$, and 0.5 CaCl$_2$, (pH 7.4, oxygenated with carbogen). The brain was then quickly extracted, and coronal slices (300 μm) were cut using a vibrating microtome (Leica-VT1000S). Slices were incubated for 10 to 12 minutes in 32°C NMDG solution before being transferred to a holding chamber filled with artificial cerebrospinal fluid (aCSF), kept at room temperature and containing (in mM) 124 NaCl, 24 NaHCO$_3$, 2.5 KCl, 1.2 NaH$_2$PO$_4$, 5 HEPES, 12.5 glucose, 2 MgSO$_4$, and 2 CaCl$_2$, (pH 7.4, oxygenated with carbogen). For electrophysiology, slices were transferred to a submerged recording chamber perfused with aCSF (3 ml/min flow rate, 30°C). Patch-clamp recordings were obtained from LS GABAergic (VGAT-positive) neurons expressing the fluorescent marker GFP and visualized using a 40× water immersion objective on an upright Olympus microscope. Recordings were performed and analyzed using an Axon Multiclamp 700B amplifier and the Clampfit10 software (Molecular Devices, San Jose, California, United States of America). The intrapipette solution contained (in mM) 126 K-gluconate, 4 KCl, 10 HEPES, 4 Mg2ATP, 0.3 Na$_2$GTP, and 10 PO-Creatine, adjusted to pH 7.25 with KOH (272 mosm). Pipette resistance was 4 to 6 MΩ.

## Calcium imaging analysis

Calcium imaging videos were analyzed using the Miniscope Analysis pipeline (https://github.com/etterguillaume/MiniscopeAnalysis) as described previously [47]. Nonrigid motion correction was applied using NoRMCorre [119], and videos were spatially downsampled (3×)

before concatenation. Calcium traces were extracted using CNMFe [120] using the following parameters: gSig = 3 pixels (width of Gaussian kernel), gSiz = 20 pixels (approximate neuron diameter), background_model = "ring," spatial_algorithm = "hals," min_corr = 0.8 (minimum pixel correlation threshold), min_PNR = 8 (minimum peak-to-noise ratio threshold). After extraction, cells and traces were visually inspected, and cells with low signal to noise ratio were removed. Raw calcium traces were filtered to remove high-frequency fluctuations and binarized (normalized amplitude > 2 SD and the first-order derivative > 0) [47].

Mouse position and head orientation was tracked using a DeepLabCut [121,122] model trained on several mouse pose markers including head centroid (for position) and nose tip (used to compute head direction angle; see below). Velocity was extracted by dividing Δd/Δt where $d$ is the distance and $t$ is time and subsequently smoothed using a Gaussian filter with sigma = 33 ms to remove movement artifacts. Acceleration was computed by differentiating (Matlab function *diff*) velocity. Head direction was computed as the angle between the vertical axis and the line formed by the head centroid and nose tip. Location data were interpolated to calcium imaging sampling frequency using linear interpolation. Head directions were interpolated using the same method, but direction data were first unwrapped, interpolated, and converted back into radians.

## Activity and bursting index

Following the binarization of raw calcium traces, we compute the probability of a neuron to be active P(A) using the following formula:

$$P(A) = \frac{time\ active}{total\ time}$$

The bursting index was computed as the probability of a cell being active, given it was already in an active state, or P(Active, t | Active, t-1). The activity index is computed as the probability of a cell becoming active when it is in an inactive state, or P(Active, t | Inactive, t-1).

## Spatial modulation

Before processing, input variables were binned (Position in open field: $17 \times 17$ bins of $3 \times 3$ cm; Position in linear track: 34 bins of 3 cm; velocity, min = 2.5 cm/s, max = 30 cm/s, 20 bins; acceleration: min = −2 cm/s$^2$, max = +2 cm/s$^2$, 20 bins; head direction: 40 bins of 9°). Velocity was smoothed over 30 frames to reduce discontinuities. For head direction and positional information, all frames where velocity was <5 cm/s were excluded. The probability of being active P(A) informs on the activity rate of a neuron. Next, we compute the probability of spending time in a given bin $i$ (spatial, velocity, acceleration, or head direction bin).

$$P(S_i) = \frac{time\ in\ bin\ i}{total\ time}$$

Next, we compute the joint probability:

$$P(S_i \cap A) = \frac{time\ active\ while\ in\ bin\ i}{total\ time}$$

And the probability that a cell is active given the animal is in a bin:

$$P(A|S_i) = \frac{time\ active\ while\ in\ bin\ i}{time\ in\ state}$$

After plotting the binarized activity in space, we perform $n = 1,000$ random circular permutations. Circular permutations are used to remove the temporal relationship between neuronal activity and behavior, but it preserves the temporal structure of calcium transients and therefore leads to more conservative results (as opposed to complete randomization of every data point, which inflates the significance value of results). Because shuffled surrogates were not systematically normally distributed, we used a nonparametric approach where the $p$-value corresponds to the number of data points from the shuffled distribution that are greater than the actual data for each bin, divided by the number of permutations [123]. A threshold of $p < 0.01$ is used to determine the significant data points that make up a significant place field. In our case, for a cell to be considered spatially modulated, at least one of the spatial bins must contain activity that is significantly nonrandom. To compute within-session stability, the halfway point in the recording was determined by taking the timestamps where half of the total number of binarized events for the session was reached. Tuning maps were computed for both the first and second half of the session using the methodology described above. A Gaussian smoothing filter ($\sigma = 3$) was used to smooth the tuning maps before computing a correlation between the two. A within-session correlation value of $>0.5$ was considered a stable field.

## Calculating mutual information content

MI is used to describe the amount of information about one variable (spatial location, head direction, velocity, or acceleration) through the observation of neuronal activity and was calculated using the following formula:

$$MI = \sum_{i=1}^{M} \sum_{j=1}^{2} P\left(S_i \cap A_j\right) \times log2\left(\frac{P(S_i \cap A_j)}{P(S_i) \times P(A_j)}\right)$$

where $M$ is the total number of possible behavioral states, $P(S_i \cap A_j)$ is the joint probability of the animal being in bin $i$ concurrently with activity level $j$. As we are using binarized activity, $j$ can only be active or inactive. MI was calculated using 30 bootstrapped surrogates using 50 percent of the data (randomly chosen data points) with replacement, to allow for computation of mean MI and SEM. Next, the trace was circularly shuffled in order to calculate a shuffled MI and SEM. Cells were then sorted along with the magnitude of the difference between mean MI and mean shuffled MI, and a two-way ANOVA with a $p < 0.01$ threshold was used to determine whether the difference between the 2 numbers was significant in order for the cell to be deemed significantly spatially modulated. In case the assumptions for parametric ANOVA are not met (normal distribution, variance homogeneity), the significance value was computed by taking the sum of the shuffled MI values of greater magnitude than actual MI values, divided by the number of shuffles.

## Bayesian decoding

A Bayesian decoder was used to evaluate how well LS neural activity estimated the animal's location as compared to neural activity recorded from dorsal CA1 and CA3. We used decoding methods specifically adapted for calcium imaging data with binarized calcium transients [47]. Using only epochs with velocity $>5$ cm/s, a training dataset was generated using 50% of the data. The remaining 50% of the data was used for testing. Decoding efficiency (as measured by a decoding error and a decoding agreement score as outlined below) was calculated using 30 bootstrapped surrogates using randomly chosen data points with replacement, to allow for computation of mean MI and SEM. In order to allow for fair comparison between recordings with different numbers of cells recorded, this approach was done using either 40, 60, 80, or 100

randomly chosen cells with replacement for each bootstrapped sample. The posterior probability density function, which is the probability that an animal is in a particular bin $S_i$, given the neural activity A, is calculated using the following equation:

$$P(S_i|A) = \frac{P(A|S_i) \times P(S_i)}{P(A)}$$

P(S|A) is the posterior probability distribution of states given neuronal activity. No prior assumption is made about the location of the mouse on the linear track; P(S) is kept uniform to attribute equal probability for each location. Every neuron is assumed to be independent of each other. To construct the tuning curves from multiple neurons, we use the following equation:

$$P(S|A) = \prod_{k=1}^{N} \frac{P(A_k|S) \times P(S)}{P(A_k)}$$

With $P(S|A)$ a vector of a posteriori behavioral states and $N$ corresponding to the number of neurons used. For every neuron $k$, the tuning curves are constructed, and corresponding posterior location probability can be derived.

To reconstruct the position of the mouse, we will consider the location associated with the maximum a posteriori:

$$\hat{y} = arg\ max\ exp\left[\sum_{k=1}^{N} log(1 + \frac{P(A_k|S) \times P(S)}{P(A_k)}) - 1\right]$$

With the estimated state among all possible states S. We used a temporal filtering window in the open field of 1.5 seconds and in the linear track of 0.5 seconds to remove erratic jumps in the decoded position of the mouse. Decoding accuracy was measured as (1) decoding error and (2) decoding agreement. In 1D space, decoding error was assessed as

$$decoding\ error = |decoded\ position - actual\ position|$$

In 2D space, decoding error was assessed as

$$decoding\ error = (\sqrt{actual\ position - decoded\ position})^2$$

Decoding score was computed as

$$decoding\ score = \frac{mean\ (shuffled\ decoding\ error) - mean\ (decoding\ error)}{length\ of\ track}$$

Decoding agreement was defined as the portion of time where the exact location or head direction bin was successfully decoded:

$$decoding\ agreement = \frac{time\ points\ successfully\ decoded}{total\ time}$$

## Tracking cells across sessions

Neurons were tracked over multiple days using a probabilistic method as previously described [124]. Briefly, spatial footprints were aligned using nonrigid alignment (transformation smoothness = 2) to correct for brain tissue displacements or miniscope placement. After alignment, we considered candidate sets of cells to be the same neuron if their maximal distance was <12 μm, and the spatial correlation between sets of cells was >0.65. Next, we visually

identified candidate cell pairs across sessions and manually deleted those cell pairs that were erroneously selected (<1% of cases). We assessed the stability of the spatial representation in LS using simple field correlation, which was assessed by correlating smoothed place fields of cell pairs. To generate the null hypothesis for place fields' displacements between pairs of days, we used the place fields tuning maps but shuffled cell identities over days (30 shuffles).

## Materials availability

This study did not generate new unique reagents.

## Statistics

Statistical analyses were performed using Matlab (MathWorks) and GraphPad Prism version 6.00 (GraphPad Software, La Jolla, California, USA). All data are presented as mean ± standard error of the mean (SEM), and statistical test details are described in the corresponding results. All $t$ tests are two-tailed. Normality distribution of each group was assessed using the Shapiro–Wilk normality test, and parametric tests were used only when distributions were found normal, using Student $t$ tests, one-way ANOVA, or two-way ANOVA. For non-normal distributions, nonparametric tests Mann–Whitney or Kruskal–Wallis tests were performed and described where applicable. Significant main effects or interactions were followed up with appropriate post hoc testing using Bonferroni corrections were applicable. $p < 0.05$ was considered statistically significant, or $p < 0.01$ and described where applicable *, $p < 0.05$; **, $p < 0.01$; ***, $p < 0.001$, ****, $p < 0.0001$.

## Supporting information

**S1 Data. Excel spreadsheet containing, in separate sheets, the underlying numerical data and statistical analysis for figure panels 1F, 1G, 1H, 2D, 2E, 3B, 3C, 3D, 3E, 3H, 4C, 4D, 4E, 4F, 5B, 5D, 5F, 7C.**
(XLSX)

**S2 Data. Excel spreadsheet containing, in separate sheets, the underlying numerical data and statistical analysis for Supporting information figure panels S1B, S1H, S2C, S2D, S3C, S4B, S4D, S4E, S4F, S4G, S5E, S6A, S6B, S6C, S6D, S7B, S7C, S7D, S7E, S8A, S8B, S8C, S8D, S8E, S8F, S8G, S8H, S9B, S9C, S9D, S9E, S9H, S10D, S10E, S10F, S10G, S10H, S10I, S10J, S11D, S11E, S11F, S12A, S12B, S13A, S13B, S13C, S13D, S13E, S13F, S14A, S14B, S14C, S14D, S14E, S16B, S16C, S16D, S16E, S17A, S17B, S17C, S17D, S17E, S20C, S20D, S20E, S21A, S21B, S21C.**
(XLSX)

**S1 Fig. LS receives direct inputs from the pyramidal cells of the hippocampus.** (A) Strategy for retrograde rabies tracing in LS. (B) Mean proportion of cells found in the hippocampus for each 300 μm coronal section; two-way RM ANOVA, F(24,126) = 2.738, $p$ = 0.002, interaction effect, $n$ = 5 mice. (C) Coronal section of ventral hippocampal region, showing retrogradely labeled cell bodies in subiculum, CA1, and CA3. (D) Retrogradely labeled cell bodies in the FC. (E) Retrogradely labeled cells in the VTA. (F) Bilateral labeling of the hypothalamic region. (G) Cell bodies in the periaqueductal gray. (H) Total number of starter cells for each LS subregion, ANOVA, F(2,12) = 24.61, $p$ > 0.0001. Scale bars: C, left: 500 μm, right: 300 μm. D, 200 μm. E, 500 μm, F, 500 μm and G, 400 μm. *, $p$ < 0.05, **, $p$ < 0.01, ***, $p$ < 0.001. Test used in B, one-way ANOVA. The underlying data can be found in S2 Data. Aq, aqueduct; f, fornix; DG, dentate gyrus; FC, fasciola cinereum; HPC, hippocampus; LH, lateral hypothalamus; LS, lateral septum; LSd, dorsal lateral septum; LSi, intermediate lateral septum; LSv,

ventral lateral septum; MM, medial mammillary nucleus; mtg, mammillotegmental tract; ns, not significant; PAG, periaqueductal gray; PH, posterior hypothalamic area; Sub, subiculum; VTA, ventral tegmental area; 3V, third ventricle.
(PDF)

**S2 Fig. Histological verification of CA1 implant location.** (A) Examples for CA1 animals included in 1D spatial navigation task, with Cre-dependent GCaMP6f (green) in a CaMKIIα-Cre mouse with DAPI counterstaining (blue). (B) Corresponding SFPs from extracted cells, color coded for minimum to maximum MI value. (C) Example of approach to compute the correlation between MI and position on the x-axis, linear regression $R^2 = 0.005682$, $p = 2678$. (D) Summary of all $R^2$ values for $n = 5$ mice. The underlying data can be found in S2 Data. MI, mutual information; SPF, spatial footprint.
(PDF)

**S3 Fig. Histological verification of CA3 implant location.** (A) Coronal section of dorsal hippocampal region, showing location of GRIN lens implant. Left: animals implanted *trans*-hippocampally. Right: animals implanted with an extrahippocampal approach. (B) Coronal sections from anterior to progressively more posterior regions of the hippocampus, showing extent of damage for a 500-μm lens implant. (C) Comparison between MI and probability being active for spatially modulated cells recorded in animals with *trans*-hippocampal vs. extrahippocampal implants (MI: unpaired *t* test, t(64) = 0.0367, $p = 0.9708$; P(A): unpaired *t* test, t(51) = 0.3908, $p = 6875$; extra-hipp: $n = 22$ cells from 2 animals, *trans*-hipp: $n = 44$ cells from 3 animals). The underlying data can be found in S2 Data. GRIN, gradient refractive index; MI, mutual information; ns, not significant.
(PDF)

**S4 Fig. GABAergic cells in the LS are significantly spatially modulated in a 1D environment.** (A) Example of raw calcium fluorescence trace (top) and binarized events (green, bottom). Trace is divided into early and late recording epochs using 50% of total calcium (using the area under the curve). Inset: zoomed calcium trace with binarized traces. (B) Linear correlation between MI and probability being active (linear regression, $R^2 = 0.5006$, $p < 0.0001$, $n = 1,030$ cells from $n = 15$ mice). (C) Method for computing the MI and 95% confidence interval from bootstrapped samples (left) and 30× shuffled surrogates (right). (D) MI computed from actual traces (black) and shuffled traces (red), sorted by the magnitude of the difference between these values, two-way ANOVA, F(1029,59740) = 126.3, $p < 0.0001$ for interaction effect. Left inset: zoomed version first 20 cells. Right inset: zoomed version for 20 not significant cells. (E) Scatterplot for significance level for spatial modulation and probability being active (linear regression, $R^2 = 0.0145$, $p = 0.007$, $n = 1,030$ cells, $n = 15$ mice). (F) Group averages of actual (gray) vs. shuffled (red) MI for spatially modulated LS cells recorded on linear track (Mann–Whitney test, U = 116,394, $p < 0.0001$). (G) LS group averages of spatial cells (gray) vs. nonspatial cells (red) for split half stability (Mann–Whitney test, U = 17,743, $p < 0.0011$). (H) Examples of significantly spatially modulated cells representative for each rank, as ranked according to panel C. *, $p < 0.05$, ****, $p < 0.0001$. The underlying data can be found in S2 Data. LS, lateral septum; MI, mutual information.
(PDF)

**S5 Fig. Intrinsic properties and firing frequency in GABAergic neurons of the LS and PV-positive fast-spiking interneurons of the hippocampus.** (A) VGAT-cre mice were injected with AAVdj-Flex-GFP in LS, and patch-clamp whole cell recordings were performed from fluorescent neurons of the dorsal LS. (B) Representative photos showing the localization of Cre-dependent fluorescence and VGAT-positive neurons 2 weeks after bilateral

microinjections at LS coordinates. Scale bar: 200, 200, 10 μm, (asterisk marks the location of a patched cell). (C) Current-clamp traces from a GFP-positive LS neuron (shown in b) characterized using depolarizing current injection steps (0–200 pA). (D) Sample traces showing hyperpolarizing and depolarizing responses from a fast-spiking (PV) hippocampal interneuron. (E) Plot of mean firing frequencies in response to injected currents of increasing suprathreshold amplitudes in hippocampal PV interneurons (open circles) and LS VGAT neurons (solid squares). Current injections were 600 ms square pulses. Firing frequencies are plotted from threshold current (t) to t + 220 pA (two-way ANOVA with Bonferroni's multiple comparison test, $F_{(11,154)} = 21.74$, $p < 0.0001$ for interaction effect). *, $p < 0.05$, ***, $p < 0.001$, ****, $p < 0.0001$. The underlying data can be found in S2 Data. LS, lateral septum; PV, parvalbumin.
(PDF)

**S6 Fig. High proportion of spatially modulated cells in LS is not caused by differences in probability being active.** (A) Probability being active for all cells recorded for each group (Kruskal–Wallis, $H(3) = 502.0$, $p < 0.0001$; LS: $n = 1,030$ cells, $n = 15$ mice. CA1: $n = 1,251$ cells, $n = 5$ mice, CA3: $n = 464$ cells, $n = 6$ mice). (B) Histogram of probabilities being active for all groups (C) Histogram of MI values for all cells recorded in LS, CA1, and CA3. (D) Using a cutoff of $P(A) > 0.001$ to exclude cells of low activity levels, comparison of proportion of spatially modulated cells for each region (left, Kruskal–Wallis, $H(3) = 59.57$, $p = 0.0509$, LS: $n = 15$ mice. CA1: $n = 5$ mice, CA3: $n = 6$ mice) and average split half stability for spatially modulated cells (right, Kruskal–Wallis, $H(3) = 80.48$, $p < 0.0001$, LS: $n = 475$ cells, $n = 15$ mice, CA1: $n = 363$ cells, $n = 5$ mice, CA3: $n = 142$ cells, $n = 5$ mice). ****, $p < 0.0001$. The underlying data can be found in S2 Data. LS, lateral septum; MI, mutual information.
(PDF)

**S7 Fig. LS spatially modulated cells are significantly directionally modulated.** (A) Mouse location on day 5 of linear track training, divided in left runs (orange) and right runs (light blue) with corresponding raw calcium activity (middle, dark blue) and derived binary trace (bottom, yellow). (B) Probability of cell being active for left (orange) vs. right trajectories (blue), and corresponding MI calculated separately for left and right runs. (C) Corresponding locations where binarized activity was detected (orange for left trajectories, blue for right trajectories and black for all trajectories). (D) MI values for right vs. left trajectories ranked by magnitude of the difference between the two. Dotted line $p < 0.01$ significance level used to assess spatial modulation of cells, two-way ANOVA, $F_{(489,28420)} = 496.9$, $p < 0.0001$ for interaction effect. (E) Proportion of spatially modulated cells in LS that are right selective (blue, 22.08%), left selective (orange, 15.52%), spatially modulated but not directionally modulated (red, 7.67%), and nonspatially modulated (gray, 54.71%, left), CA1 (middle) and CA3 (right). The underlying data can be found in S2 Data. LS, lateral septum; MI, mutual information.
(PDF)

**S8 Fig. Decoding using fewer cells still leads to significant differences between septohippocampal regions and shuffled surrogates in linear track.** (A) Mean decoding error for location decoding on the linear track using LS cells computed for 30 bootstrap samples of 40, 60, 80, or 100 cells (black, each dot represents mean of an animal) compared to a decoded location using shuffled tuning maps (red, each dot represents mean of an animal), two-way RM ANOVA, $F_{(3,19)} = 571.4$, $p = 0.0024$ for main effect of shuffling, $F_{(3,19)} = 6.948$, $p = 0.0024$ for main effect of number of included cells. (B) Left: same as A, for mean decoding agreement, two-way RM ANOVA, $F_{(3,19)} = 1059$, $p < 0.0001$ for main effect of shuffling, $F_{(3,19)} = 10.06$,

$p$ = 0.0003 for main effect of number of included cells, Right: method for computing the mean decoding agreement for each bootstrapped estimate. (C) Same as A, for cells recorded from dorsal CA1. In addition to 30 bootstrapped samples of 40, 60, 80, and 100 cells, this panel includes mean decoding error using all recorded cells, two-way RM ANOVA, F(1,20) = 304.3, $p$ < 0.0001 for main effect of shuffling, F(4,20) = 2.647, $p$ = 0.0637 for main effect of number of included cells. (D) Same as A, for cells recorded from dorsal CA3, two-way RM ANOVA, F (1,9) = 42.83, $p$ = 0.0001 for main effect of shuffling, F(3,9) = 1.480, $p$ = 0.2845 for main effect of number of included cells. (E) Using only cells with P(A) > 0.001, decoding error for actual vs. shuffled dataset using 60 cells (paired $t$ tests, LS: t(5) = 0.10, $p$ = 0.001, $n$ = 6 mice; CA1: t(4) = 6.732, $p$ = 0.0025, $n$ = 5 mice; CA3: t(3) = 2.966, $p$ = 0.0592, $n$ = 4 mice). (F) Using only cells with P(A) > 0.001, mean decoding score for each region (one-way ANOVA, F(2,12) = 5.765, $p$ = 0.0167. (G) Same as E, without using a temporal smoothing window (paired $t$ tests, LS: t(6) = 10.08, $p$ < 0.0001, $n$ = 7 mice; CA1: t(4) = 7.897, $p$ = 0.0014, $n$ = 5 mice; CA3: t(3) = 2.614, $p$ = 0.0794, $n$ = 4 mice). (H) Same as F, without using a temporal smoothing window (ANOVA, F(2,13) = 5.462, $p$ = 0.0190). *, $p$ < 0.05, **, $p$ < 0.01, ***, $p$ < 0.001, ****, $p$ < 0.0001. The underlying data can be found in S2 Data. LS, lateral septum; ns, not significant.
(PDF)

**S9 Fig. GABAergic cells in the LS are significantly spatially modulated during free exploration in a 2D environment.** (A) Probability P(active | location) of an example cell to be active in a bin (3 cm) of the open field (45 or 49 cm size). (B) Significance computed from $n$ = 1,000 circular permutations, $p$ < 0.01 is considered significant. (C) Within-session stability is computed by correlation of the first half (left) and the second half (right) of the recording. (D) MI computed from actual traces (red) and shuffled traces (black), sorted by the magnitude of the difference between these values (one-way ANOVA, F(1898,110142) = 65.50, $p$ < 0.0001 for interaction effect). Left inset: zoomed version of first 20 cells. Right inset: zoomed version for 20 not significant cells. (E) Cells are ranked according to the difference between their bootstrapped mean MI value and 30 circularly shuffled surrogates. Ranks are color coded for clarity, $n$ = 1,899 cells from $N$ = 28 mice. (F) Examples of significantly spatially modulated cells with corresponding mean MI value (top) and split within-session stability (bottom) representative for each rank. (G) Computation of activity index (probability inactive to active) and bursting index (probability active to active). (H) Left: activity index for spatial vs. nonspatial cells (Mann–Whitney test, U = 342,219, $p$ < 0.0001). Right: bursting index for spatial vs. nonspatial cells (Mann–Whitney test, U = 368,276, $p$ < 0.0001). The underlying data can be found in S2 Data. LS, lateral septum; MI, mutual information; ns, not signifiicant.
(PDF)

**S10 Fig. LS spatially modulated cells do not anchor to food zones or objects.** (A) Experimental setup, with animals freely exploring the same open field for 5 consecutive days, with 2 food zones and 2 objects added in opposing corners on days 2 and 4, respectively. Representative trajectories are shown in gray. Green circles and stars represent locations of food and objects. (B) Representative example of the average number of cells active per 3-cm bin for an example mouse (baseline 1: $n$ = 132 cells; food zones: $n$ = 110 cells; baseline 2: $n$ = 118 cells; objects: $n$ = 118 cells; baseline 3: $n$ = 72 cells). (C) For the same animal, red dots are centroids of spatially modulated cells. (D) Centroids for each significantly spatially modulated cell of all animals included in analysis (baseline 1: $n$ = 730 cells, $n$ = 10 mice; food zones: $n$ = 559 cells, $n$ = 10 mice; baseline 2: $n$ = 749 cells; $n$ = 10 mice; objects: $n$ = 407 cells, $n$ = 8 mice; baseline 3: $n$ = 471 cells; $n$ = 8 mice). (E) Time spent in food zone as a percentage of total time spent in session for all animals ($n$ = 10 mice). (F) Number of visits to food zone ($n$ = 10 mice). (G)

Likelihood of cell being active within the food zone as compared to a shuffle for all cells recorded (Wilcoxon matched-pairs signed rank test, W = −3,490, $p$ = 0.6497, $n$ = 559 cells, $n$ = 10 mice). (H) Time spent in object zone as a percentage of total time spent in session for all animals ($n$ = 8 mice). (I) Number of visits to objects ($n$ = 8 mice). (J) Likelihood of a cell being active closely around the objects as compared to a shuffle (Wilcoxon matched-pairs signed rank test, W = 3,789, $p$ = 0.4251, $n$ = 407 cells, $n$ = 8 mice). The underlying data can be found in S2 Data. LS, lateral septum; ns, not significant.
(PDF)

**S11 Fig. Significantly spatially modulated cells are not centered around the reward zone in a nonmatch to place task.** (A) Schematic of T-maze nonmatch to place task, consisting of a first forced run, followed by a free run in which the previously nonvisited arm is rewarded. (B) Example trajectory of well-trained animal, with Probability (active | location) of 3 example cells using 5-cm bins. (C) For 2 example animals, red dots are centroids of spatially modulated cells. (D) Time spent in food zone as a percentage of total time spent in session for all animals ($n$ = 6 mice). (E) Number of visits to reward zone ($n$ = 6 mice). (F) Likelihood of a cell being active within the reward zone as compared to a shuffle for all cells recorded (Wilcoxon matched pairs signed rank test, W = −4,783, $p$ = 0.2358; $n$ = 365 cells from $n$ = 6 mice, day 10 of training). The underlying data can be found in S2 Data. ns, not significant.
(PDF)

**S12 Fig. Differences in spatial coding not due to inclusion of pseudo-silent cells.** (A) Probability of being active for spatially modulated cells (Kruskal–Wallis, H(3) = 295.0, $p < 0.0001$; Dunn's multiple comparisons test; LS: $n$ = 718 spatial cells from $n$ = 28 mice; dCA1: $n$ = 323 spatial cells, $n$ = 6 mice; dCA3: $n$ = 138 spatial cells, $n$ = 7 mice). (B) Left: using an activity cut-off of P(A) > 0.001, proportion of spatial cells per animal (one-way ANOVA, F(2,39) = 0.9849, $p$ = 0.3826; Tukey's multiple comparisons test; LS: $n$ = 1,889 cells from $n$ = 28 mice; dCA1: $n$ = 1,017 cells, $n$ = 6 mice; dCA3: $n$ = 521 cells, $n$ = 8 mice). Middle: within-session stability for spatial cells (Kruskal–Wallis, H(3) = 89.52, $p < 0.0001$; Dunn's multiple comparisons test; LS: $n$ = 718 spatial cells from $n$ = 28 mice; dCA1: $n$ = 323 spatial cells, $n$ = 6 mice; dCA3: $n$ = 138 spatial cells, $n$ = 7 mice). Right: mean dispersion for spatial cells (Kruskal–Wallis, H(3) = 155.5, $p < 0.0001$; Dunn's multiple comparisons test; LS: $n$ = 718 spatial cells from $n$ = 28 mice; dCA1: $n$ = 323 spatial cells, $n$ = 6 mice; dCA3: $n$ = 138 spatial cells, $n$ = 7 mice). *, $p < 0.05$, **, $p < 0.01$, ***, $p < 0.001$, ****, $p < 0.0001$. The underlying data can be found in S2 Data. LS, lateral septum.
(PDF)

**S13 Fig. Decoding using fewer cells still leads to significant differences between septohippocampal regions and shuffled surrogates in 2D environment.** (A) Mean decoding error for location decoding in the open field using LS cells computed for 30 bootstrapped samples of 40, 60, 80, and 100 cells (black, each dot represents mean of an animal) compared to a decoded location using shuffled tuning maps (red, each dot represents mean of an animal; two-way RM ANOVA, F(1,33) = 174.9, $p < 0.0001$ for main effect of shuffling, F(3,33) = 0.2041, $p$ = 0.893 for main effect of number of included cells. (B) Left: same as A, for mean decoding agreement (two-way RM ANOVA, F(1,33) = 132.8, $p < 0.0001$ for main effect of shuffling, F(3,33) = 1.132, $p$ = 0.3505 for main effect of number of included cells). Right: method for computing the mean decoding agreement for each bootstrap estimate. (C) Same as A, for cells recorded from dorsal CA1. In addition to 30 bootstrap samples of 40, 60, 80, and 100 cells, panel includes mean decoding error using all recorded cells (two-way RM ANOVA, F(1,17) = 649.0, $p < 0.0001$ for main effect of shuffling, F(4,17) = 0.7948, $p$ = 0.5447 for main effect of number

of included cells) (D) Same as A, for cells recorded from dorsal CA3 (two-way RM ANOVA, $F_{(1,11)} = 44.13$, $p < 0.0001$ for main effect of shuffling, $F_{(3,11)} = 17.99$, $p = 0.9078$ for main effect of number of included cells). (E) Effect of temporal filtering on decoding error in the open field for LS (two-way RM ANOVA, $F_{(1,25)} = 123.7$, $p < 0.0001$ for main effect of shuffling, $F_{(4,25)} = 0.1169$, $p = 0.9753$ for main effect of number of included cells. (F) Comparison of mean decoding error using $P(A) > 0.001$ activity cutoff for cell selection vs. without such cutoff using 80 cells (LS: $n = 8$ mice, CA1: $n = 4$ mice, CA3: $n = 4$ mice; two-way RM ANOVA, $F_{(1,12)} = 1.374$, $p = 2.639$ for main effect of activity cutoff, $F_{(2,12)} = 1.023$, $p = 0.3889$, for main effect of structure). *, $p < 0.05$, **, $p < 0.01$, ***, $p < 0.001$, ****, $p < 0.0001$. Test used in A–F, two-way RM ANOVA. The underlying data can be found in S2 Data. LS, lateral septum; ns, not significant.
(PDF)

**S14 Fig. LS place code over days.** (A) Within-session stability for each spatially modulated cell recorded cells in LS (Kruskal–Wallis, $H(4) = 8.921$, $p = 0.0304$; Dunn's multiple comparisons test; day 1, $n = 181$; day 2, $n = 172$; day 3, $n = 178$; day 8, $n = 209$, $n = 5$ mice). (B) Average MI for each day for all recorded cells (Kruskal–Wallis, $H(4) = 13.27$, $p = 0.0041$; Dunn's multiple comparisons test; day 1, $n = 562$; day 2, $n = 600$; day 3, $n = 477$; day 8, $n = 492$, $n = 5$ mice). (C) Proportion of cells for tuning map correlation (Friedman test, $\chi^2(2) = 0.400$, $p = 0.9537$, significant cell pairs only, day 1–2, $n = 158$; day 2–3, $n = 122$, day 3–8, $n = 129$). (D) Proportion of stable cells (tuning map correlation $> 0.3$) for each progressive day correlation (one-way ANOVA, $F_{(3,16)} = 1.457$, $p = 0.2637$; significant cell pairs only, day 1–2, $n = 158$; day 2–3, $n = 122$, day 3–8, $n = 129$). (E) Proportion of spatially modulated cells per day (day 1, $n = 562$; day 2, $n = 600$; day 3, $n = 477$; day 8, $n = 492$, $n = 5$ mice). *, $p < 0.05$, **, $p < 0.01$. The underlying data can be found in S2 Data. LS, lateral septum; MI, mutual information; ns, not significant.
(PDF)

**S15 Fig. Direction encoding by LS cells.** (A) Examples of significantly modulated cells. The polar plot indicates the probability of the cell being active as a function of the animal's head direction. Black lines indicate p(active | bin); red lines indicate 95% upper and lower percentile; blue lines indicate the normalized time spent in each direction. MI calculated using 9° bins. (B) Same as A, for nonmodulated cells. (C) Trajectories (gray) with binarized activity superimposed, color coded from beginning to end of the recording for representative cells shown in A. (D) Same as C, but for example, cells shown in B. (E) Examples of significantly directionally modulated cells with corresponding mean MI value (top) and within-session stability (bottom) representative for each rank. The underlying data can be found in S2 Data. LS, lateral septum; MI, mutual information.
(PDF)

**S16 Fig. Stability of direction and self-motion encoding over short and longer timescales.**
(A) Tuning plots for a stable directionally modulated cell over days, using a similar setup as described in Fig 4. Tuning map correlation indicated at the bottom in red. (B) Significant tuning map correlation for aligned cell pairs (black) vs. shuffled pairs (red) for progressive days for LS (day 1–2, $n = 161$ cells; day 2–3, $n = 119$ cells; day 3–8, $n = 110$ cells; $n = 5$ mice) and dorsal CA1 (day 1–2, $n = 149$ cells; day 2–3, $n = 102$ cells; day 3–8, $n = 90$ cells; $n = 3$ mice). (C) Significant direction map correlations for cells found on all days for LS ($n = 29$ cells) and CA1 ($n = 24$ cells). (D) Same as B, for velocity tuning in LS (day 1–2, $n = 110$ cells; day 2–3, $n = 73$ cells; day 3–8, $n = 73$ cells; $n = 5$ mice) and dorsal CA1 (day 1–2, $n = 109$ cells; day 2–3, $n = 72$ cells; day 3–8, $n = 61$ cells; $n = 3$ mice). (E) Same as B, for acceleration tuning in LS (day 1–2,

$n$ = 73 cells; day 2–3, $n$ = 47 cells; day 3–8, $n$ = 43 cells; $n$ = 5 mice) and dorsal CA1 (day 1–2, $n$ = 45 cells; day 2–3, $n$ = 40 cells; day 3–8, $n$ = 35 cells; $n$ = 3 mice) ****, $p$ < 0.0001. Test used in B, D, E: two-way ANOVA, with Sidak's multiple comparisons test. The underlying data can be found in S2 Data. LS, lateral septum; ns, not significant.
(PDF)

**S17 Fig. Direction, speed, and acceleration encoding in dorsal hippocampus.** (A) Left: distribution of MI (left) and $p$-values (right) for direction for dCA1, dCA3, and LS. Right: MI (bits) per binarized event, for all cells recorded from each region (Kruskal–Wallis, H(3) = 351.2, $p$ < 0.0001; Dunn's multiple comparisons test; LS: $n$ = 1,230 cells, $n$ = 19 animals, CA1: $n$ = 677 cells, $n$ = 4 animals, CA3: $n$ = 546 cells, $n$ = 7 animals). (B) Same as A, for velocity (Kruskal–Wallis, H(3) = 325.3, $p$ < 0.0001; Dunn's multiple comparisons test) (C) Same as B, for acceleration (Kruskal–Wallis, H(3) = 262.2, $p$ < 0.0001; Dunn's multiple comparisons test). (D) Left: proportion of cells that are significantly modulated by only one modality (gray), 2 modalities (yellow), 3 (red) or all 4 of the investigated variables (black) for dCA1 ($n$ = 677 cells, $n$ = 4 mice). Right: absolute proportion of cells modulated by any combination of variables. E) Same as D, for dCA3 ($n$ = 546 cells, $N$ = 7 mice). *, $p$ < 0.05, ****, $p$ < 0.0001. The underlying data can be found in S2 Data. A, acceleration; D, direction; LS, lateral septum; MI, mutual information; S, spatial coding; V, velocity.
(PDF)

**S18 Fig. LS GABAergic neurons project to the MS, hypothalamus, and VTA.** (A) Injection of anterograde, Cre-dependent eYFP (green) viral tracing in VGAT-Cre mouse LS. (B) Injection site in LSi/LSd with some cell bodies labeled in LSi (green, eYFP; blue, DAPI counterstaining). (C) eYFP-positive fibers in the MS, (D) hypothalamus, and (E) VTA. (F) Coronal hippocampal section shows no anterograde labeling of the hippocampal formation, either dorsal or (G) ventral. Scale bars: B, 500 μm; C, 500 μm; D, 500 μm; E, 500 μm; F, 500 μm; G, 700 μm. The underlying data can be found in S2 Data. DB, diagonal band; DG, dentate gyrus; LS, lateral septum; LSd, dorsal lateral septum; LSi, intermediate lateral septum; LSv, ventral lateral septum; MS, medial septum; S, subiculum; TA, ventral tegmental area; 3V, third ventricle.
(PDF)

**S19 Fig. Extent of primary injection sites in dorsal CA1 and CA3.** (A) Coronal sections showing expression of tdTom (red) in dorsal CA1 along anterior to posterior axis. (B) Same as A, for primary injections in dorsal CA3. Scale bars: 500 μm for all sections. The underlying data can be found in S2 Data.
(PDF)

**S20 Fig. Anterograde transsynaptic tracing shows that CA1 and CA3 project preferentially to dorsal LS.** (A) Primary AAV1 injections in CA1 and CA3, using the same injection strategy as described in Fig 6A. (B) Coronal section showing expression pattern at different anterior–posterior levels of the LS, with (red) tdTOM-positive CA3 projections to LS and eYFP-positive second-order transduction in LS (left). Bottom schematic: an overview of the approximate bregma level of coronal slices shown. Bottom right: zoomed versions showing tdTom-positive hippocampal afferents, eYFP-positive LS cell bodies, and merge. (C) Total eYFP-positive cells counted at the level of LS following transsynaptic tracer injection in dCA1 vs. dCA3 (dCA1-LS, $N$ = 3 mice; dCA3-LS, $N$ = 5 mice). (D) Total eYFP-positive cells along the dorsal–ventral and anterior–posterior axes observed in LS for dorsal CA3 injection and (E) for dorsal CA1 injection. Scale bars: B, top: 500 μm; bottom: 500 μm. C, all overview images, 500 μm, all zoomed images, 50 μm. The underlying data can be found in S2 Data. LS, lateral septum; LSd, dorsal lateral septum; LSi, intermediate lateral septum; LSv, ventral lateral septum; ns, not

significant.
(PDF)

**S21 Fig. No changes in cell activity along dorsal–ventral or anterior–posterior axis of the LS.** (A) Probability of being active for each cell along the anterior–posterior axis (left), medial–lateral axis (middle), and dorsal–ventral axis (right). All cells were recorded during free exploration in the open field (linear regression, AP: $R^2 = 2.014 \times 10^4$, $p = 0.5611$; ML: $R^2 = 1.012 \times 10^4$, $p = 0.6803$; DV: $R^2 = 2.717 \times 10^4$, $p = 0.6803$; $n = 1,679$ cells, $n = 24$ mice). (B) Same as A, but for activity index as described in S5 Fig (linear regression, AP: $R^2 = 2.691 \times 10^4$, $p = 0.5017$; ML: $R^2 = 3.287 \times 10^5$, $p = 0.8144$; DV: $R^2 = 3.254 \times 10^4$, $p = 0.4601$; $n = 1,679$ cells, $n = 24$ mice). (C) Same as for A, but for bursting index (linear regression, AP: $R^2 = 7.085 \times 10^4$, $p = 0.2757$; ML: $R^2 = 7.457 \times 10^5$, $p = 0.7237$; DV: $R^2 = 3.731 \times 10^3$, $p = 0.0123$; $n = 1,679$ cells, $n = 24$ mice). The underlying data can be found in S2 Data. LS, lateral septum.
(PDF)

## Acknowledgments

We thank Fernanda Sosa for help with behavioral recording experiments, training of the animals, immunohistological experiments, and quantifications; Alexandra T. Keinath for valuable comments on a previous version of the manuscript; and Ke Cui for help with colony maintenance and mice perfusions. We also thank Daniel Aharoni (UCLA) for guidance in using the UCLA miniscope and Bruno Rivard for help building the miniscopes. The present study used the services of the Molecular and Cellular Microscopy Platform at the DHRC.

## Author Contributions

**Conceptualization:** Suzanne van der Veldt, Frédéric Manseau, Sylvain Williams.

**Data curation:** Suzanne van der Veldt.

**Formal analysis:** Suzanne van der Veldt.

**Funding acquisition:** Suzanne van der Veldt, Sylvain Williams.

**Investigation:** Suzanne van der Veldt, Guillaume Etter, Coralie-Anne Mosser.

**Methodology:** Suzanne van der Veldt, Guillaume Etter.

**Project administration:** Suzanne van der Veldt, Sylvain Williams.

**Resources:** Sylvain Williams.

**Software:** Suzanne van der Veldt, Guillaume Etter.

**Supervision:** Sylvain Williams.

**Visualization:** Suzanne van der Veldt, Frédéric Manseau.

**Writing – original draft:** Suzanne van der Veldt.

**Writing – review & editing:** Guillaume Etter, Sylvain Williams.

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
