## [Editor Report · Decision Letter 0]

23 Mar 2021

Dear Dr Williams, 

Thank you for submitting your manuscript entitled "Conjunctive spatial and idiothetic codes are topographically organized in the GABAergic cells of the lateral septum" for consideration as a Research Article by PLOS Biology.

Your manuscript has now been evaluated by the PLOS Biology editorial staff, as well as by an academic editor with relevant expertise, and I am writing to let you know that we would like to send your submission out for external peer review. Please accept my apologies for the delay in sending this decision to you.

Please re-submit your manuscript within two working days, i.e. by Mar 25 2021 11:59PM.

Kind regards,

Gabriel Gasque, Ph.D.,

Senior Editor

PLOS Biology

---

## [Decision Letter · Decision Letter 1]

6 May 2021

Dear Dr Williams,

Thank you very much for submitting your manuscript "Conjunctive spatial and idiothetic codes are topographically organized in the GABAergic cells of the lateral septum" for consideration as a Research Article at PLOS Biology. Your manuscript has been evaluated by the PLOS Biology editors, by an Academic Editor with relevant expertise, and by four independent reviewers. You will note that reviewer 1, Felix Leroy, has revealed his identity. Please accept my apologies for the delay in communicating the decision below to you.

The reviews of your manuscript are appended below. You will see that the reviewers find the work potentially interesting. However, based on their specific comments and following discussion with the Academic Editor, I regret that we cannot accept the current version of the manuscript for publication. We remain interested in your study and we would be willing to consider resubmission of a comprehensively revised version that thoroughly addresses all the reviewers' comments. We cannot make any decision about publication until we have seen the revised manuscript and your response to the reviewers' comments. Your revised manuscript would be sent for further evaluation by the reviewers.

Having discussed the reviews with the Academic Editor, we have decided that reviewer 2’s point 1 (“In order to confirm the findings, LS activity should also be measured electrophyisiologically (e.g. tetrode recording).”) and reviewer 3’s point about “targeted inactivation studies” are out-of-scope. Thus, we will not press for these additional data. However, you would need to strengthen your conclusions elsewhere with additional data and analyses, where requested, to satisfy the reviewers, including reviewer 3’ fourth main concern (“I would strongly advise the author to re-do tracing studies while targeting more proximal portion of the CA1 too.”)

We appreciate that these requests represent a great deal of extra work, and we are willing to relax our standard revision time to allow you six months to revise your manuscript.We expect to receive your revised manuscript within 6 months.

**IMPORTANT - SUBMITTING YOUR REVISION**

*Resubmission Checklist*

*Published Peer Review*

*PLOS Data Policy*

*Blot and Gel Data Policy*

Sincerely,

Gabriel Gasque

Senior Editor

PLOS Biology

ggasque@plos.org

REVIEWS:

Reviewer #1: Van der Veldt and colleagues imaged calcium transients in the lateral septum and hippocampus during spatial navigation. Thanks to their numerous recordings at different LS locations, they effectively investigate the topographical organization of the spatial and idiothetic codes within LS. The manuscript is well-written and provides by far the best account of the downstream copy of the hippocampal spatial code to LS. Overall, this is an excellent manuscript and I highly recommend it for publication. I have a few minor comments below. 

Line 141: we thus observed a significantly higher proportion of place cells in LS as compared to CA1 and CA3 (Kruskall-Wallis, H(3) = 6.102, p = .0298, Fig 1F).

Did the author group CA1 and CA3 for this comparison? It's unclear from the text or figure what is being compared.

Line 143: We found that even though within-session stability of place cells was significantly different between LS, dCA1, and dCA3, stability in the LS was not significantly different than in dorsal CA1 (Kruskall-Wallis, H(3) = 8.922, p = .0116, Fig 1G).

Is LS different from CA3? The figure only indicates a difference between CA1 and CA3.

Line 204: In order to assess spatial coding during free exploration in a two-dimensional environment, we recorded Ca2+

Superscript for 2+ is missing.

Line 2010: we did not observe any increased number of place cells with fields around food zones, nor did we observe overrepresentation of objects or walls.

Provide a quantification of the time spent eating and exploring the objects. A simple explanation for not seeing any modulation around the food zones and objects could be that the mice failed to eat or explore. In addition, did you look at the calcium event rate in the rewarding zones?

Line 349: We observed that dCA1-LS preferentially targets the LH, and dCA3-LS targets the hypothalamus more broadly as well as the nuclei of the medial zone.

Perhaps you could compare this result to the proposed topographical projection from hippocampus to hypothalamus (Risold and Swanson, Science 1996)

I couldn't find where Supplementary figure 9 is described in the main text.

Line 358: functional projections to medial septum neurons.

I think the word "functional" does not apply here and should be removed since the data presented is solely neuronal tracing.

Line 966: (A) Dual-viral injection strategy same as described for S10 Fig.

Do you mean Fig. 5A? In addition, the identical color code (red, blue and green) used for the bregma coordinates (C) and the dorso-ventral localization in LS (F) is slightly confusing.

Line 385: Both dorsal dCA1 and dCA3 injections resulted in the strongest transfection of LS cell bodies in dorsal regions compared to intermediate and ventral LS (S10D-F Fig).

I agree with the conclusion but since the proportion of transfected cells will depend on where the Cre-dependent virus is injected (AP: 0.5; ML: 0.35; DV: -2.6 = dLS), I would prefer if you mentioned this possible caveat. Overall, if you want to conclude on the density of hippocampal innervation to LS, the pattern of TdTomato-labelled fibers is probably more indicative.

Fig 6C: typo in the y-axis labels: "Proporton"

Line 609: injected in the LS of CaMKIIα-Cre mice.

Line 353: We used an AAV-Flex-Synaptag...

Why not inject a non-Cre-dependent synaptag virus in WT mice or the FLEX synaptag following trans-synaptic Cre-delivery from hippocampus to LS? Ad in the main text that the virus was injected in a CaMKII-Cre mouse. In this and a few other occasions it's a little unclear in what region the viruses were injected.

Line 497: The question arises of what the function of could be having a region downstream

Correct the sentence.

Reviewer #2: In this manuscript, van der Veldt and colleagues recorded the activity of GABAgic neurons in the LS using in vivo calcium imaging technique. The authors report that the activity of the cells is modulated by place, speed, acceleration, and direction. The information content of these cells is comparable with those of hippocampal cells. The distribution of cells showing these properties formed gradients along A-P and D-V axes of the LS. Long-term tracking of LS neuronal activity revealed that a subset of LS cells has stable spatial representation over days. Using viral tracing they showed that LS neurons receive CA3 and CA1 excitatory input and send projections to the hypothalamus and medial septum.

Overall, this study presents multiple new findings, and the manuscript is well-written. However, interpretation of spatial distribution of place/kinematic cells in the LS seems somewhat difficult, because the invasiveness of endoscopic imaging. In my opinion, the manuscript is still suitable for publication in PLOS Biology, after the authors have addressed the following comments and questions.

1. Spatial organization of information in the LS (Fig. 6) is potentially very interesting, however, insertion of the GRIN lens significantly damages the local circuits when imaging from ventral part of the LS. Because the LS GABAgic cells are highly interconnected, imaging with GRIN lens would perturb their activity. In order to confirm the findings, LS activity should also be measured electrophyisiologically (e.g. tetrode recording).

2. The authors reported that, contrary to the previous report, the LS place cells did not show overrepresentation of reward zone (Fig. S6). This discrepancy is likely due to the task demand: According to a previous report (Dupret et al., 2010 Nat Neurosci, 8 995), overrepresented activity around reward zone in the CA1 was observed only when the reward was hidden and the animals should navigate themselves to the reward zone using their spatial memory. Therefore, the following points should be clarified:

1. Was food or foodzone visible for the animals? 

2. Did the CA1 cells show overrepresentation of reward in the same experimental condition?

3. Fig. 3B: The authors found that some LS cells show highly stable place representation. What percentage of the cells shows such a high stability? It would be more informative if graph showing tuning map correlation vs cell number is added. 

4. In addition to spatial representation, is self-motion information represented stably in LS cells?

5. Fig. 3G: It seems slightly higher correlation for shorter interval in CA1 than in LS. Tuning map correlation for 1-day-interval as well as for day1-8 should be quantified.

6. Page 20 line 497: The authors discuss the possibility that the LS cells use sparse code. However, I could not find any data showing that activity of LS cells is more sparse than that of hippocampal cells. As shown in Fig. 2D, LS cells are more active than CA1/CA3 cells, and Fig. 2E showed that place field size is larger in LS than in CA1/CA3. These data are opposed to sparse coding.

Minor Comments:

1. Fig. 3A: GCAMPF6f -> GCAMP6f

Reviewer #3: Summary:

VanderVeldt et al present here a very interesting dataset of 1000+ fast-spiking neurons recorded with miniscopes covering a large extent of the lateral septum (LS) of mice freely behaving on a linear track or in an open field. They analyze how the activity of these neurons is modulated by spatial, directional and self-motion correlates and compare their findings with another dataset they recorded in similar conditions from the CA1 and CA3 regions of the hippocampus. Spatial, directional and self-motion coding had already been reported multiple times in the LS. Yet, the reported proportions of LS neurons modulated by these correlates vary a lot from on study to the other. VanderVeldt et al hypothesize that these discrepancies stem from an anatomical topographic organization of spatial coding. This hypothesis had already partially been explored by previous studies which did not find neither medio-lateral (Wirtshafet and Wilson 2020) nor dorso-ventral (e.g., Takamura et al., 2006) organization, yet they did not have access to a dataset as large as the one presented here. 

I agree that that the use of miniscopes is indeed well appropriate to thoroughly test the existence of a topographic organization of spatial coding in LS, as well as to reconciliate apparent discrepancies within the literature. However, the results on topography only come in figure 6 and most of the findings presented prior to figure 6 are not very novel. Most of the data presented here aims to compare coding in LS fast-spiking GABAergic cells and CA1 pyramidal cells. I have several concerns related to the methods used by the authors to compare these very different populations that I will outline below. Moreover, the novelty of these findings is not striking, given that Wirtshafet and Wilson had already done so in a study published in elife in 2020.

In addition to their imaging studies, VanderVeldt et al present results from tracing studies to demonstrate a topographical organization of CA1-to-LS projections that they claim to correlate with the coding organization. Their main conclusion being that LS is a relay between the hippocampus and other subcortical structures such as the medial septum (MS) and the hypothalamus. I am not sure how much new results is provided here compared to previous studies. Moreover, the method here are not always appropriate to reach the conclusions outlined in the main text - see detailed concerns below. This is especially problematic when it comes to their main conclusion on LS being a relay. This conclusion cannot be reached based on the results presented here. It could be tested with targeted inactivation studies, which would be a whole other set of experiments.

Recommendation: 

I agree that the question of coding in LS neurons (and its topographical functional organization) is very important to understand the neural mechanism supporting information processing, especially with regards to the putative role of LS as a hippocampal relay. I appreciate the interesting and rich dataset provided here, as well as the attempt at using tracing studies to supplemental their claims based on the miniscope results. However, I am not sure whether the findings of VanderVeldt offer- as for now - a substantive increase in our knowledge on that subject (except for figure 6). The promise of "functionally characterize hippocampal connectivity of LS" is very appealing but I do not think that the work presented here fulfill it. I was also put off by many imprecisions both in the text and in the figures, with incomplete legends or even a supplementary figure 9 that is not referred to within the text. My opinion is that this work will need major rewrite/rework/refocus in order to reach the quality and novelty of findings expected from articles published in PlosBiology. I am listing below my principal major and minor concerns, yet, given the high number of imprecisions, this is not an exhaustive list.

Major concern:

- As stated in the summary above, my main concern is on the lack of novelty of the results presented here. The main advantage of VanderVeldt et al dataset over previous studies is the use of miniscope which allows to image simultaneously a large number of neurons, depict their precise anatomical location and morphology, as well as their activity in freely behaving animals, over several consecutive days. I don't think that this rich dataset was exploited to its fullest here, especially when reporting/comparing results with CA1 imaging. I urge the authors to highlight better, especially in their results portions, how what they present here is new and contrast with what has already been published.

- My second main concern is about comparing two very different populations: LS fast spiking neurons and CA1 pyramidal cells. First of all, this point is not made clear enough in the text. Second, it is problematic for the analyses because: 

o 1) Many CA1 pyramidal cells are silent or pseudo-silent in a given environment. This would influence many of the analyses presented here (e.g., modulation for a specific correlate, decoder). Reports of spatial/directional/self-motion coding on cells with insufficient activity rate are meaningless. Likewise, it is not right to compare decoders based on set of cells with different percentage of silent cells. The authors must set a threshold of activity (electrophysiologic studies have set such a threshold as 100 spikes/10 mins) for cells to be analyzed. ALL analyses, comparison of percentages and use of neurons for decoding must be re-done while considering only active cells (or comparable percentage of active cells). 

o 2) Even when comparing only active cells, LS fast spiking neurons have very different rate compared to CA1 or CA3 pyramidal neurons. The authors must check to which extend rate influence what they measure (e.g., scoring methods, decoder, etc.). If they observe a rate effect, they must control the effect on scoring methods by calculating "information per binarized unit of activity" or by downsampling. Likewise, they must control the effect on decoder by proceeding to their random selection of cells to decode from on cells presenting similar rate or presenting downsampled rate when comparing CA1/CA3 and LS. In SF6, I noticed a higher number of spatial cells coding for area that are infrequently visited (centre). This raises concerns as to the fact that authors may include cells with too low threshold of activity.

- My third main concern is that the number of mice in CA3 is too low to reach any conclusion. I think that for this result to be part of the main figures, the authors should present results from at least 3 mice. This is especially of concern given that I am suspecting that the imaging procedure may compromise the integrity of the hippocampal circuits by cutting intrahippocampal connective fibers. I would very much appreciate if the authors could ease these concerns by presenting complete histology of hippocampal recordings. The integrity of the hippocampal circuits is essential to assess spatial information there and to meaningfully compare it with LS spatial information. 

- My fourth main concern is that I have doubt about whether the authors considered the fact that CA1 pyramidal cells present a strong heterogeneity in spatial coding along the dorso-ventral and proximo-distal axis in CA1 (Henriksen et al 2010). I urge the authors to incorporate information on proximo-distal location of CA1 pyramidal cells when reporting analyses on spatial modulation. To my knowledge, only 2 examples of CA1 field of view are reported for now. I would like to see the others reported too. Likewise, injection in the CA1 for the tracing studies seems limited to the most distal portion of CA1 - an area which is known to have less spatial information. I would strongly advise the author to re-do tracing studies while targeting more proximal portion of the CA1 too.

- My fifth main concern is about the remapping analyses and is linked to an incomplete use -in my view- of the rich information provided by miniscopes. First: given the fact that the authors can follow the activity of individual cells across days, I do not understand why they estimate remapping based on the whole population. I would very much prefer to see a cell-by-cell analyses instead that would allow us to distinguish between 2 main scenarios for the LS population: 1) part of the population remain stable while the other remap and 2) all cells remap equally. Second: cell-by-cell analyses should allow the authors to address the crucial difference between rate and phase remapping. As stated above, we know that many CA1 cells are silent (or become more silent when the environment gets more familiar). Differencing between rate and phase remapping is thus essential to understand the dynamics between CA1 and LS.

Minor concerns:

- It may be a matter of taste, but I am not sure that there is a consensus on the fact that directional information is purely idiothetic - according to me it is not.

- I would prefer if the authors could use the term spatially modulated cells rather than place cells. The definition of a place cell being narrower than what they present here.

- Careful when formulating that LS is the main subcortical output of the hippocampus. I don't think it is true when it comes to the subiculum or the dentate gyrus, even when limiting oneself to the dorsal portion. Also, the long-range GABAergic projection from CA1 to the MS should be mentioned already in the introduction. In general, hippocampal is used as a synonym for CA1-CA3 in the text. It should not.

- References: L81: I would cite Swanson and Cowan, 1979; Risold and Swanson, 1997; L116: I would cite Muller and Kubie 1987. L159: Souza 2018 is in the text but not in the reference list.

- L119 (and several times after): the authors make a point about the fact that: in contrast [to the hippocampal formation] LS has been implicated in behaviors that may require a stable representation of space over time, including context-reward, contextual fear conditioning and spatial learning and memory. This line of argument is flawed as a large body of studies have shown that the hippocampal formation is involved in all these behaviours. This argument is repeated in the discussion. This is problematic: CA1 (and the parahippocampal region) has been demonstrated to encode information about goal/reward.

- Fig 1B, S1 (and subsequent histology/tracing images): I would very much prefer if number of infected cells were reported as estimated percentages rather than in raw numbers. As it is, the authors should report % infected cells/DAPI. However, it would be even better to proceed to additional Immunohistochemistry (IHC) staining to report percentages by cell type. For instance, to prove that they indeed target pyramids in the hippocampus. I am a bit puzzle by the use of rabies (a quite heavy procedure) to merely show projections from CA1-3 to the LS. 

- On S1: what is the percentage of starter cells? Could the authors please report the % of GFP+ only and mCherry+ only cells in the LS? That would give an idea of the efficiency of the rabies strategy. Did the authors verify that it actually infected only GABA+ cells and that there was no leakage?

- Please use the terms infected and infection (or transduction) instead of transfected and transfection.

- Fig 1B, S1: given that labelling is inequal depending on anatomical location, it is insufficient to only report the distance to the bregma if the plan is to spatially map inputs onto LS cells. It would be more accurate to take into account all parameters, including M/L, D/V and proximo-distal axis, as well as ideally cell type if further immunohistochemistry analyses are possible. Given that the results are pooled from 5 mice, I would appreciate if the authors could provide data/mouse on the graphs.

- L 171: "within-session stability of place cells was significantly different between LS, dCA1, and dCA3, stability in the LS was not significantly" Please reformulate.

- A few points on the decoder: 1) Do the result change without the temporal filtering of the decode position? The filtering introduces temporal correlations that could boost performance but not be related to the information content of the population. 2) The procedure for performance calculation is not illustrated clearly: the authors say they train on 50% of the data, but they do not say if the rest is used for testing, and how many times they do the train-test split. Please clarify. If it is not what the authors have already done, I would suggest a cross validation procedure (10-fold cross validation or leave-one-out cross validation are the most used in decoding approaches as far as I know). 3) Strong issue about rate (see main concern)

- 20 cm decoding error on a 100 cm track is not very accurate. I would adjust the text to reflect that. Maybe this is due to the inclusion of many silent cells in the CA1 assembly?

- 3- Bursting index: I would suggest for clarity the probability of two consecutive active bins to be denoted with P(A,t|A,t-1) instead of P(A|A). A matter of taste, more than anything.

- Fig 1H and 3G are not informative with the present scales

- L240: no increase at food reward: not surprising in a novel environment. The degree of novelty/familiarity of the environment is improperly neglected in the analyses when this would have been very valuable information. When appropriate, please report number of exposures and time spent in each environment. Make sure to compare cells of animals receiving the same amount of exposure. This greatly influences spatial information content.

- The difference in stability results between linear track and open field are not addressed.

- Fig 3: 1) I would appreciate a picture evidencing virus infection in the LS. 2) Fig 3H "Tuning map correlation between day 1 and day 8" is based on only one animal for CA1. Is that the whole extends of the data? If so that seems insufficient to reach a conclusion on that point. 

- Remapping analyses: I would have been curious to know more about partial remapping (near the border of the environment for example). The dataset presented here could be exploited to answer this interesting question. However, this analysis is not as important in view of the others. 

- Head direction (HD) analyses: is it head direction or heading? Please be more specific in methods. Also detail how velocity and acceleration are calculated. When showing polar plot for HD, please add a central circle to represent (hopefully) uniform head direction of the animal during that session. Also note in the discussion that retrosplenial cortex and parasubiculum also present large proportion of directionally modulated cells.

- Fig 5: As it is this figure is not fully supporting the authors claims. After reading the study that initially published the AAV1 virus used here, I could not see reports of any 2nd order neurons labelled. Therefore, I am not sure whether the virus can actually spread further than the 1st order neuron. Could the authors please discuss this point. This is problematic in light of S10 where the same technique is used to label direct inputs. It is not clear to me how they can claim that there is no second order labelling in S10. Furthermore, I am unsure that based on the evidence provided her, the author can reach the conclusion that there is direct disynaptic pathway from principal cells of the hippocampus to the medial septum. I also would like for the authors to show all the injected area rather than a few examples. 

- It was not clear to me whether the same AAV1 Cre was used for 5D. Incidentally, I'm not sure that this is the best way to prove that these are synaptic connections. The method used her will indeed show that there are putative boutons in these areas, but to prove that these are synaptic connections, one need to do an IHC with another marker of the presynaptic terminal and a post-synaptic marker… Or to test this functionally with a ChR2 for instance. All in all, I would very much appreciate if the authors could highlight the novelty of their results here compared to what is already known on these projections. (also true for S9)

- Fig 6 "thus it is likely that this information is directly inherited from the hippocampus." I do not think that the data presented here can support this statement. Inactivation studies would be more appropriate to test this.

Reviewer #4: PlosReview

The authors used 1-photon mini-scopes to record from neurons in the mouse lateral septum and assessed the coding characteristics of these neurons in terms of egocentric and allocentric spatial and behavioral parameters. In addition, they used transsynaptic viral tracing to characterize the efferent projection of LS neurons that receive direct hippocampal connections, showing connections with medial septal and hypothalamic regions not directly connected with hippocampus. The central finding is that LS neurons represents spatial, speed, and directional information in an anatomically-organized fashion in a manner corresponding somewhat to the mapping from D-V axis of hippocampus onto LS. The GABAergic LS spatial cells were quite comparable to hippocampal "place" cells.

Overall critique:

This is a very interesting paper that adds significantly to the literature on the inheritance, in extrahippocampal structures, of spatial correlates from the hippocampus. The technological achievement is laudable, and the analysis has been competently executed. I think the paper will be of interest to a wide audience. I have only quite minor comments and suggestions.

There are a few grammatical errors which can be left to the editors to correct.

"We found that even though within-session stability of place cells was significantly different between LS, dCA1, and dCA3, stability in the LS was not significantly different than in dorsal CA1 (Kruskall-Wallis, H(3) = 8.922, p = .0116, Figure 1G)." Sentence appears to be contradictory. Also, poor syntax: "different than" should be "different from".

Figures: Text is generally too small, and the figures are mostly too complex. For example, in Figure 1 it would be better to split out A, B & C to a separate figure. Caption for Fig 1 lacks an explanation of part E - jumps from D to F

"LS integrates direction, velocity, and acceleration information" - This is perhaps a bit of an overstatement. LS cells express such information, but it is not clear in what sense they integrate it, nor is it even clear what that would mean. Integration of velocity gives speed, integration of speed gives relative position, but I don't know what integrating direction would give.

Care needs to be taken in interpreting 'directional modulation' in open field situations where walls restrict access of the animals to certain orientations in certain positions.

Authors might wish to point out that this is not the only case of GABAergic neurons inheriting some spatial specificity from hippocampal pyramidal cells. Indeed, interneurons within CA1 show a very comparable effect - higher firing rate, greater spatial dispersion but overall comparable information per second (Maurer et al., 2006, J. Neurosci. 26:13485).

Although it seems pretty likely that the observed correlates are indeed inherited from hippocampus, the authors should at least address the possibility that it comes from other brain regions, that project to LS, of which there are many.

References to the literature on the change in the properties of hippocampal cells along the DV axis would be appropriate in the discussion section (Jung et al., J. Nrsc. 1994, 14, 7347; Kjelstrup et al., Science 2008, 321, 140).

Typo: "The question arises of what the function of could be having a region downstream" - move the second "of" to after "be".

"Thus, through this process of coincidence detection and recurrent inhibition, the spatial map could be compressed to be represented by a much lower number of cells. As such, the LS could accurately convey hippocampal information to downstream regions using a much sparser signal, thereby allowing for increased metabolic efficiency (Olshausen and Field, 2004) and more effective associative learning (Isely et al., 520 2010; Palm and Sommer, 2006; Zetsche). " … It's a matter of opinion perhaps, but I think that this section is a bit incorrectly phrased. The idea of compression appears to be correct and has been brought up previously in the context of hippocampal projection to subiculum, for example. But compression means that the output code is actually more 'distributed' not sparser, i.e., in LS, a greater proportion of the neurons is activated at a given location, the firing rates are higher, and so the net information per second transferred is the same. Such a non-sparse code is more appropriate for data transmission rather than associative storage. There is increased metabolic demand at the single cell level, but considerably fewer cells can be used to transmit the same information.

---

## [Editor Report · Decision Letter 2]

20 Jul 2021

Dear Dr Williams,

Thank you for submitting your revised Research Article entitled "Conjunctive spatial and idiothetic codes are topographically organized in the GABAergic cells of the lateral septum" for publication in PLOS Biology. I have now obtained discussed your new version with other staff editors and with the Academic Editor as well. I am pleased to tell you that we will probably accept this manuscript for publication, provided you satisfactorily address the following point:

We would like you to acknowledge more in the discussion the potential caveat of circuit damage because of the use of GRIN lens. This may be particularly an issue when imaging in the CA3 area. The Academic Editor disagrees with your statement that damage of CA1 should not affect CA3 because CA1 lesion will disrupt grid cells in the entorhinal cortex which in turn is expected to influence spatial activity to some degree in CA3.

We would also like you to consider making your title more accessible to the broad readership of PLOS Biology, and thus, suggest the following:

"Spatial and self-motion information is topographically organized in the GABAergic cells of the lateral septum." 

We would be happy to work with you on an alternative, if you think our recommendation is not accurate or misrepresents your findings. 

Please also make sure to address the data and other policy-related requests listed below my signature. 

We expect to receive your revised manuscript within two weeks. 

*Published Peer Review History*

*Early Version*

Sincerely,

Gabriel Gasque, Ph.D.,

Senior Editor,

ggasque@plos.org,

PLOS Biology

ETHICS STATEMENT:

Please include within your manuscript the ID number of the protocol approved by the McGill University and Douglas Hospital Research Centre Animal Use and Care Committee.

DATA POLICY:

Note that we do not require all raw data. Rather, we ask for all individual quantitative observations that underlie the data summarized in the figures and results of your paper. For an example see here: http://www.plosbiology.org/article/info%3Adoi%2F10.1371%2Fjournal.pbio.1001908#s5

These data can be made available in one of the following forms:

Regardless of the method selected, please ensure that you provide the individual numerical values that underlie the summary data displayed in the following figure panels: Figures 1F-H, 2DE, 3B-E, 3H, 4C-F, 5BDF, 7C, S1BH, S2CD, S3C, S4BD-G, S5E, S6A-D, S7B-E, S8A-H, S9B-E, S9H, S10D-J, S11D-F, S12AB, S13A-F, S14A-E, S16B-E, S17A-E, S20C-E, and S21A-C.

Please also ensure that each figure legend in your manuscript includes information on where the underlying data can be found and that your supplemental data file/s has/have a legend.

---

## [Editor Report · Decision Letter 3]

2 Aug 2021

Dear Dr Williams,

On behalf of my colleagues and the Academic Editor, Jozsef Csicsvari, I am pleased to say that we can in principle offer to publish your Research Article "Conjunctive spatial and self-motion codes are topographically organized in the GABAergic cells of the lateral septum" in PLOS Biology, provided you address any remaining formatting and reporting issues. These will be detailed in an email that will follow this letter and that you will usually receive within 2-3 business days, during which time no action is required from you. Please note that we will not be able to formally accept your manuscript and schedule it for publication until you have made the required changes.

***IMPORTANT:

1) As you address the remaining formatting and reporting issues, please ensure that each figure legend in your manuscript includes information on where the underlying data can be found. You can write, for example, “The numerical data used in this figure are included in Data S1/S2”

2) I modified your Data Availability Statement in the submission system to specify where in github you code can be found. Please verify it and let me know if you disagree or have questions or concerns.

PRESS

Sincerely, 

Gabriel Gasque, Ph.D. 

Senior Editor 

PLOS Biology

ggasque@plos.org